# Dopamine signaling enriched striatal gene set predicts striatal dopamine synthesis and physiological activity in vivo

Leonardo Sportelli [1,2,16], Daniel P. Eisenberg [3,16], Roberta Passiatore[2], Enrico D'Ambrosio [2,4], Linda A. Antonucci[2], Jasmine S. Bettina[3], Qiang Chen[1], Aaron L. Goldman[1], Michael D. Gregory [3], Kira Griffiths[4,5], Thomas M. Hyde [1,6,7], Joel E. Kleinman [1,7], Antonio F. Pardiñas [8], Madhur Parihar [1], Teresa Popolizio[9], Antonio Rampino[2,10], Joo Heon Shin [1], Mattia Veronese [11,12], William S. Ulrich[1], Caroline F. Zink[13], Alessandro Bertolino[2,10], Oliver D. Howes [4], Karen F. Berman[3], Daniel R. Weinberger [1,6,7,14,15] ✉ & Giulio Pergola [1,2,7] ✉

The polygenic architecture of schizophrenia implicates several molecular pathways involved in synaptic function. However, it is unclear how polygenic risk funnels through these pathways to translate into syndromic illness. Using tensor decomposition, we analyze gene co-expression in the caudate nucleus, hippocampus, and dorsolateral prefrontal cortex of post-mortem brain samples from 358 individuals. We identify a set of genes predominantly expressed in the caudate nucleus and associated with both clinical state and genetic risk for schizophrenia that shows dopaminergic selectivity. A higher polygenic risk score for schizophrenia parsed by this set of genes predicts greater dopamine synthesis in the striatum and greater striatal activation during reward anticipation. These results translate dopamine-linked genetic risk variation into in vivo neurochemical and hemodynamic phenotypes in the striatum that have long been implicated in the pathophysiology of schizophrenia.

Schizophrenia (SCZ) is a mental illness with complex heritability and polygenic architecture[1]. The largest genome-wide association study (GWAS) to date has identified an extensive set of potential SCZ risk genes converging on the synaptic biology of central nervous system

neurons[2]. To the extent that the downstream consequences of diverse risk alleles might affect shared biological functions, genetic risk for SCZ is likely best understood in the context of molecular ensembles, rather than at a single gene level. This perspective puts gene

[1]Lieber Institute for Brain Development, Johns Hopkins Medical Campus, Baltimore, MD, USA. [2]Group of Psychiatric Neuroscience, Department of Translational Biomedicine and Neuroscience, University of Bari Aldo Moro, Bari, Italy. [3]Clinical and Translational Neuroscience Branch, National Institute of Mental Health, Intramural Research Program, NIH, DHHS, Bethesda, MD, USA. [4]Department of Psychosis Studies, Institute of Psychiatry, Psychology and Neuroscience, King's College London, London SE5 8AF, UK. [5]Holmusk Technologies, New York, NY, USA. [6]Department of Neurology, Johns Hopkins University School of Medicine, Baltimore, MD, USA. [7]Department of Psychiatry and Behavioral Sciences, Johns Hopkins University School of Medicine, Baltimore, MD, USA. [8]MRC Centre for Neuropsychiatric Genetics and Genomics, Cardiff University, Cardiff, UK. [9]Radiology Department, IRCCS Ospedale Casa Sollievo della Sofferenza, San Giovanni Rotondo, Italy. [10]Azienda Ospedaliero Universitaria Consorziale Policlinico, Bari, Italy. [11]Department of Information Engineering, University of Padua, Padua, Italy. [12]Department of Neuroimaging, Institute of Psychiatry, Psychology and Neuroscience, King's College London, London SE5 8AF, UK. [13]Baltimore Research and Education Foundation, Baltimore, MD, USA. [14]Department of Neuroscience, Johns Hopkins University School of Medicine, Baltimore, MD, USA. [15]Department of Genetic Medicine, Johns Hopkins University School of Medicine, Baltimore, MD, USA. [16]These authors contributed equally: Leonardo Sportelli, Daniel P. Eisenberg. ✉e-mail: Daniel.Weinberger@libd.org; Giulio.Pergola@libd.org

co-expression at the forefront of investigating genetic risk convergence as an instrumental approach to model the effect of many variants on interconnected genetic systems and, ultimately, downstream neurochemical and neural functioning[3–5].

A large body of evidence implicates synaptic dysfunction and neurotransmission across several key brain circuits that bridge the striatum, the dorsolateral prefrontal cortex (DLPFC), and the hippocampus (HP) as key pathological mechanisms in SCZ[6–9]. As such, understanding gene co-expression across multiple brain regions may reveal how broad genetic variation translates into an increased risk of illness[1]. This translation is especially important as polygenic approaches usually lack biological characterization.

There is also a large body of evidence for dopamine involvement in SCZ, including the emergence of psychotic symptoms (e.g., hallucinations and delusions) following the administration of pro-dopaminergic agents and therapeutic antipsychotic effects elicited by dopamine-blocking drugs targeting $D_2$ receptors[10]. In the $D_2$-rich striatum where illness-related dysfunction has been observed, positron emission tomography (PET) studies have found an array of dopamine-system disturbances in SCZ suggesting increased dopaminergic drive from mesencephalic synaptic terminals, including elevated presynaptic dopamine synthesis[7,11–14]. There is also evidence that individuals at clinical risk for SCZ, e.g., with subthreshold psychotic symptoms, as well as first-degree relatives, show a similar pattern of elevated striatal presynaptic dopamine synthesis capacity[15,16], which may be enhanced with progression to frank illness[17]. Importantly, striatal dopamine synthesis shows heterogeneity across patients with SCZ[18], particularly in treatment-resistant individuals, who have demonstrated synthesis capacity decreases[19]. Different mechanisms may be at play in treatment-resistant patients[20]. Recent evidence from post-mortem human caudate nucleus (CN) has revealed that decreased expression of the short (predominantly presynaptic autoreceptor) isoform of the $D_2$ dopamine receptor gene *DRD2*−and not the long (predominantly postsynaptic) isoform−may be a causative mechanism for association of the SCZ GWAS risk allele mapped to the *DRD2* locus[19]. By identifying diminished expression of the inhibitory $D_2$ presynaptic autoreceptor as one potential mechanism of SCZ risk, this work further implicates exaggerated presynaptic dopamine activity in pathogenesis[21], consistent with earlier work associating a single-nucleotide polymorphism (SNP) with differential *DRD2* splicing, striatal dopamine D2 signaling, and prefrontal and striatal activity during working memory[22,23].

Functional magnetic resonance imaging (fMRI) studies have reported altered brain activity in patients with SCZ while performing dopamine-dependent reward processing tasks, possibly arising from synaptic dysfunction and neurotransmission dysregulation[24,25]. Moreover, anticipatory striatal activation during reward task performance has been shown to be a heritable trait[26] ($h^2 = 0.20$–$0.73$). Therefore, genetic investigations may help better define important connections between this phenotype and dopamine-relevant SCZ risk molecular factors. In light of this and because dopamine dysfunction in SCZ generally appears to have at least in part a genetic basis[21,27–29], we hypothesized that a SCZ-related genetically driven increase of striatal presynaptic dopamine synthesis might be reflected functionally in an increase of striatal fMRI activation during reward anticipation at least in neurotypical individuals.

While most of the same genes are expressed across brain regions, mRNA expression patterns vary consistently with differing functions subserved at a system level. A widely used approach to analyze gene co-expression patterns is a combination of graph theory and clustering, such as in the popular weighted gene co-expression network analysis[30], which has been extensively applied to transcriptomics of SCZ patients as well as neurotypical controls[3,27,28,31–34]. This approach, however, has important limitations in its handling of higher-dimensional data, particularly in accounting for the multiplicity of co-expression contexts across brain regions and cell types. These aspects are crucial to capture the biological reality in which different tissues, cells, and molecular pathways share common genes. Another class of co-expression detection methods called sparse decomposition of arrays (SDA) circumvents these limitations[35]. SDA is based on singular value decomposition, a family of techniques that includes independent component analysis (ICA) and principal component analysis (PCA). SDA is able to effectively identify relationships between genes in multi-tissue experiments[35]. By decomposing a 3D Array (also called a "Tensor") with dimensions representing individuals, genes, and tissues, respectively, into several latent components (or factors), SDA captures major directions of variation in the dataset. This approach identifies components that uncover functional biology[35,36] and outperforms other co-expression detection strategies in the identification of functionally related and co-regulated groups of genes[37].

Using SDA and two independent post-mortem brain samples, we investigated human RNA sequencing data from three brain regions prominently implicated in SCZ, i.e., CN, HP, and DLPFC (Fig. 1). We sought to identify gene sets enriched both for genes differentially expressed in SCZ and for genes associated with SCZ genetic risk. Focusing on gene sets with convergence of illness state and illness risk in neurotypical brain avoids epiphenomena related to drug treatment in patient samples and the same directionality of effects supports genetic risk inferences.

We identified a co-expression component in the SDA data that meets these criteria and is especially enriched for dopamine function genes. We then aimed to evaluate whether this component specifically translates into SCZ-relevant brain functional correlates in vivo. To that end, we studied striatal dopamine synthesis capacity determined via PET in both neurotypical controls (NC) and patients with SCZ and obtained corroborative evidence in an independent replication dataset. We then measured brain physiological activation during reward anticipation with fMRI in two independent neurotypical cohorts performing different reward tasks. We sought to translate dopamine-linked gene sets in the post-mortem brain involved in manifest illness and illness risk into neurochemical and neurofunctional outcomes in the living human brain concordant with known SCZ-associated phenotypes.

## Results
### Gene co-expression analysis
From the SDA of post-mortem CN, HP, and DLPFC tissue from our discovery cohort (Table 1) we obtained 69 robust components not associated with confounding variables. Supplementary Data 1 and 2 report the output of SDA as well as the association with biological covariates and technical confounders and summary information regarding the number of genes included in each component.

When comparing samples from NC and individuals with SCZ, two of 69 filtered components (C80: 2,497 genes; C109: 1,211 genes; see Supplementary Data 2 for component gene membership) were associated with diagnosis (C80: $F_{[1,210]} = 11.4$, $p = 0.0009$, $p_{[FDR]} = 0.038$, $\eta^2 = 0.05$; C109: $F_{[1,210]} = 10.9$, $p = 0.001$, $p_{[FDR]} = 0.039$, $\eta^2 = 0.03$) (Fig. 2a). To identify SCZ-associated components more likely linked to pathogenic biology rather than treatment history or other factors, we additionally tested these components across samples for association with SCZ genetic risk before proceeding with further analyses. Only the SDA component C80 was also significantly associated with SCZ polygenic risk score (PRS), a measure of overall cumulative risk burden, in a diagnosis-consistent direction (see Methods for PRS computation; C80: $t_{[93]} = 1.67$, one-tailed $p = 0.048$, partial $R^2 = 0.03$; C109: $t_{[93]} = -1.2$, one-tailed $p = 0.11$, partial $R^2 = 0.015$) (Fig. 2a). Patients with SCZ had greater C80 scores and, consistently, healthy controls with greater SCZ PRS had relatively greater C80 scores.

Biological characterization of this component showed enrichment for SCZ, major depressive disorder (MDD) and attention deficit

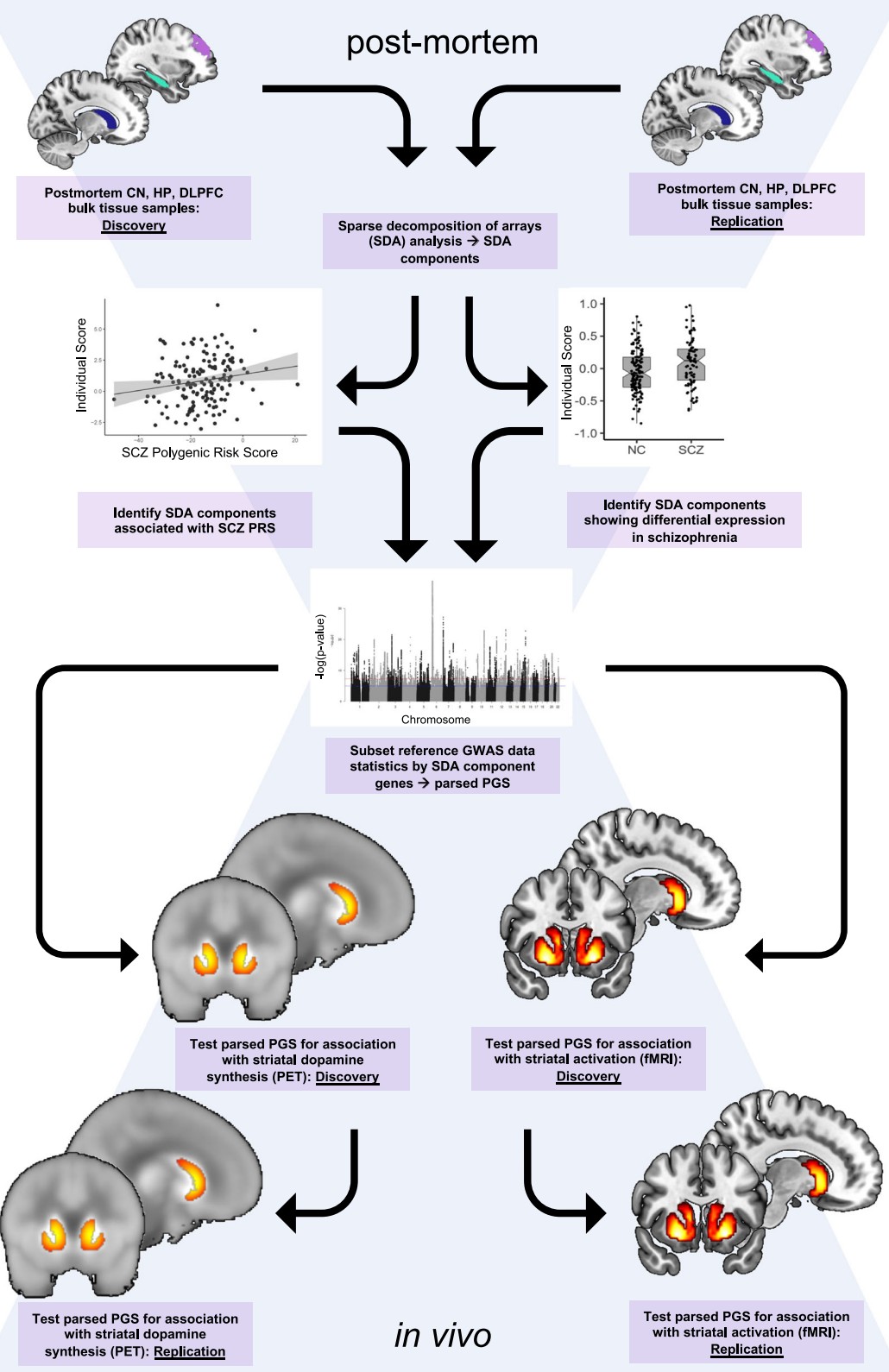

**Fig. 1** | Graphic summary of the study design.

hyperactivity disorder (ADHD) risk genes (Fig. 2b, c) as well as SCZ differentially expressed genes (DEGs) previously observed in the CN[21] and in the DLPFC, differentially methylated genes (DMG; i.e., genes proximal to regions enriched in CpG islands differentially methylated in SCZ compared to healthy controls) and loss of function intolerant

genes (all empirical $p < 0.05$; Fig. 2b). Moreover, we used Multi-marker Analysis of GenoMic Annotation (MAGMA)[38] and H-MAGMA[39], which leverages chromatin accessibility datasets, to perform a gene-set enrichment analysis for pathology-specific GWAS variants and found that the association with SCZ risk of the variants falling within or

## Table 1 | Demographics

| Modality | Cohort | Diagnosis (NC/SCZ) | N | Age (years ± SD) | Sex (N female) | Ancestry (EUR/AA) | Genes |
|---|---|---|---|---|---|---|---|
| Post-mortem data (CN, HP, DLPFC) | Discovery | NC | 154 | 44.9 ± 16.9 | 46 | 70/84 | 22,356 |
| | | SCZ | 84 | 49.4 ± 16.5 | 24 | 39/45 | |
| | Replication | NC | 120 | 59.2 ± 9.6 | 30 | 120/0 | 20,475 |
| [$^{18}$F]-FDOPA PET | Discovery | NC | 65 | 29 ± 9 | 31 | 65/0 | - |
| | | SCZ | 20 | 29 ± 8 | 3 | 20/0 | |
| | Replication | NC | 150 | 35 ± 11 | 75 | 150/0 | - |
| Reward fMRI | Discovery | NC | 86 | 32 ± 6 | 47 | 86/0 | - |
| | Replication | NC | 55 | 26 ± 6 | 34 | 55/0 | - |

Demographics of cohorts used in the gene co-expression (upper rows) and neuroimaging (lower rows) analyses are tabulated. Neuroimaging samples are those with both genetic and imaging data after ancestry stratification. Source data are provided as a Source Data file.

CN Caudate Nucleus bulk tissue data, HP hippocampus bulk tissue data, DLPFC dorsolateral prefrontal cortex bulk tissue data, NC Neurotypical controls, SCZ Patients with schizophrenia, AA African American, EUR European.

regulating (based on chromatin interactions) C80 genes was greater than that in the remaining sets ($p_{[FDR]} < 0.05$; Fig. 2b). Interestingly, this technique showed greater specificity to SCZ, as there was no consistency across MAGMA, H-MAGMA, and GWAS variant analyses with MDD and ADHD results (Fig. 2b). Further exploratory genetic risk association and biological characterization analyses of the other identified components are reported in the Supplementary Methods and Supplementary Figs. 1a, 6. The broader landscape of co-expression components highlights further potential pathways of interest linked to SCZ risk.

To determine which tissue contributed more to the interindividual variation within a given component, we evaluated the tissue score matrix obtained by SDA, which represents the covariance between the overall gene expression derived from one tissue and the component identified. Using a threshold of |0.5| (as previously reported by SDA developers[35]) in the tissue loading matrix, we found the C80 component to be most active in the CN (Fig. 2b). Accordingly, cell specificity analysis suggested a highly significant preponderance of medium spiny neurons (MSNs) and dopaminergic terminals ($p_{[FDR]} = 1.9 \times 10^{-57}$; Fig. 2d), consistent with CN localization. Gene ontology analysis (Fig. 3a, Supplementary Fig. 1b, c and Supplementary Data 3) characterized C80 as a predominantly synaptic component (133 genes, fold enrichment = 2.3, $p_{[FDR]} = 1 \times 10^{-18}$) with both pre (128 genes, fold enrichment = 1.9, $p_{[FDR]} = 2.2 \times 10^{-12}$) and postsynaptic specializations (103 genes, fold enrichment = 2, $p_{[FDR]} = 8 \times 10^{-12}$, Fig. 3a). The Kyoto Encyclopedia of Genes and Genomes (KEGG) pathway analysis showed enrichment for dopaminergic, GABAergic, glutamatergic, and cholinergic synapses, all characteristic of the CN (Fig. 3b and Supplementary Data 3).

To follow up on these network-level findings suggesting a role for this component in dopaminergic neurotransmission, we investigated the membership of C80 for dopamine receptor and synthesis genes. C80 included the dopamine $D_2$ receptor gene DRD2 but not DRD1, along with tyrosine hydroxylase (TH) and DOPA decarboxylase (DDC) genes, necessary for presynaptic dopamine biosynthesis. We found that C80 was positively correlated to DDC expression ($t_{[201]} = 5.3$, $p = 2.9 \times 10^{-7}$, partial $R^2 = 0.12$) and negatively correlated to DRD2 expression ($t_{[201]} = -3.01$, $p = 0.003$, partial $R^2 = 0.04$) in the CN after covarying for biological and technical confounders (see Methods for details). Because DDC catalyzes the last committed step of dopamine synthesis and $D_2$ receptor signaling inhibits dopamine synthesis, these results are consistent with greater dopamine synthesis capacity in individuals with greater C80, who also have a higher polygenic risk for SCZ. Greater dopamine synthesis may thus be expected to be associated with decreased DRD2 expression in this context.

Interestingly, when restricting the analysis to only healthy individuals, we also found that C80 negatively correlated with DRD2 expression in the DLPFC ($t_{[218]} = -2.1$; $p = 0.04$; partial $R^2 = 0.02$).

We further leveraged transcript-level information to disentangle to which extent the DRD2 expression variance was related to the short or long isoform in the CN (see Methods for details), and found a significant association with the long isoform expression in a consistent direction with previous gene-level analyses (DRD2 short isoform transcript: $t_{[183]} = -1.65$, $p = 0.1$, partial $R^2 = 0.014$; DRD2 long isoform transcript: $t_{[183]} = -2.2$, $p = 0.029$, partial $R^2 = 0.025$). An even stronger transcript divergence was found with only healthy individuals (DRD2 short isoform transcript: $t_{[116]} = -1.38$, $p = 0.16$, partial $R^2 = 0.016$; DRD2 long isoform transcript: $t_{[116]} = -2.8$, $p = 0.006$, partial $R^2 = 0.06$). Due to the MSN enrichment and selective presence of DRD2 compared with DRD1 in this component, we also examined this component's membership for the 29 most preferentially expressed genes in each MSN class identified by Tran, Maynard[40] in the nucleus accumbens, with specific focus to the D1_A and D2_A clusters as they represented the largest D1-MSN (67%) and D2-MSN (87%) subclasses, respectively. We assessed the statistical significance of these intersections via permutation tests (see Methods for details). Interestingly, 17 out of 29 genes were shared between C80 and D2_A (including PENK, enkephalin typically expressed by indirect pathway MSNs[41]; empirical $p < 1 \times 10^{-4}$) and only 8 out 29 were shared with D1_A (with the exclusion of DRD1 and PDYN typically expressed by direct pathway MSNs; empirical $p = 0.09$), suggesting that the $D_2$-expressing neuronal population may contribute more to the clustering observed in C80 (Fig. 3c).

Finally, to replicate our findings, we applied SDA to the Genotype-Tissue Expression (GTEx) dataset (https://gtexportal.org/home/)[42]. RNA-seq data were available for 120 NC across CN, DLPFC, and HP (see Table 1 for demographics). This replication analysis yielded 84 components which we used to assess the replication of the 69 LIBD components (see Supplementary Data 1 for SDA output). We assessed the Jaccard Index (JI), representing the overlap between gene sets, and the correlation of component-specific gene loadings as replication measures (see Methods for details). The former revealed more statistically significant replicated components (JI:62 vs. gene loading:34; empirical $p < 0.001$; Fig. 3d and Supplementary Fig. 2a). Indeed, most filtered components were replicated in GTEx (90%; Fig. 3d) with a median JI = 0.13 in the framework of a gene universe overlap between the LIBD and GTEx dataset of JI = 0.67. Interestingly, C80 was among the only four replicated components out of 69 in which the associated GTEx component (C18) was consistently found using both JI and gene loading (JI = 0.19; gene loading $R^2 = 0.19$; empirical $p < 0.001$; Fig. 3d and Supplementary Fig. 2a). Moreover, the GTEx C18 component was again mainly active in the CN with a similar neurobiological profile by cell specificity, gene ontology, and KEGG pathway analyses and a similar enrichment for dopaminergic synapse (33 genes; fold enrichment = 1.7; $p_{[FDR]} = 0.01$) (Supplementary Fig. 2b–d and Supplementary Data 3). Accordingly, we replicated the association between C18 component loadings and DRD2 expression in the CN with effect

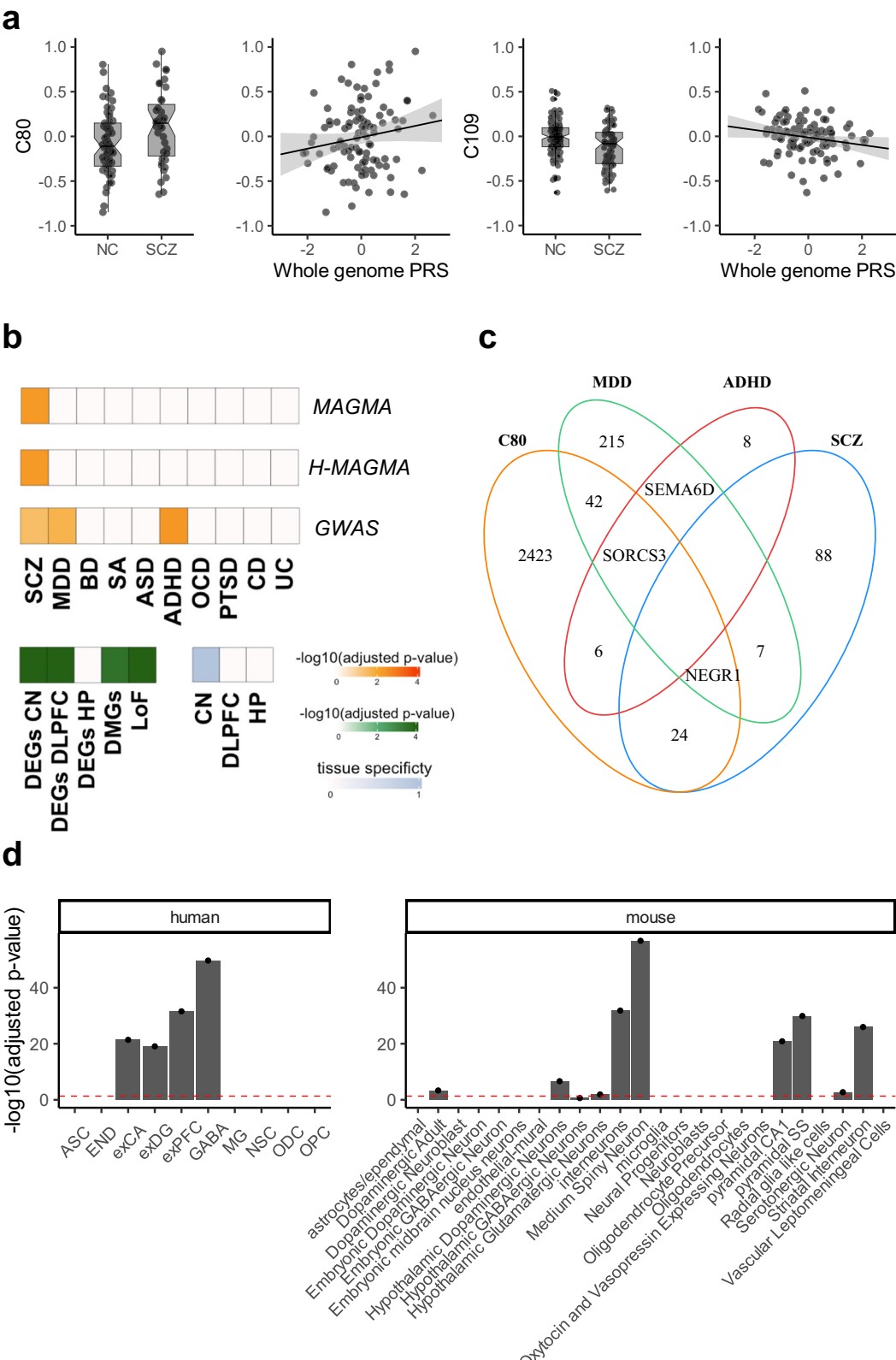

directions consistent with our discovery C80 component ($t_{[94]} = -2.6$; $p = 0.01$). We did not find a significant association with either *DRD2* long or short isoform transcripts (*DRD2* short isoform: $t_{[116]} = -1.9$, $p = 0.056$, partial $R^2 = 0.03$; *DRD2* long isoform: $t_{[116]} = -1$, $p = 0.3$, partial $R^2 = 0.008$; see Methods for details). A transcript-level tensor

decomposition might be best suited to capture the variance at this fine-grained biological resolution.

In summary, we identified replicable co-expression patterns relative to the dopaminergic neurotransmission in a completely independent dataset of neurotypical individuals. The list of genes within

**Fig. 2 | Sparse decomposition of arrays (SDA) component characterization. a** Notched box plots show SDA component C80 and C109 scores for post-mortem data samples in SCZ and NC groups (n = 229 individuals; 145 NC and 84 SCZ). These were the only components showing a significant group effect. Group medians (horizontal line), 95% confidence intervals (notches), interquartile range (box edges), and whiskers (25th/75th percentiles or extrema) are shown. The scatter plot demonstrates SDA component C80 and C109 scores as a function of polygenic risk for schizophrenia and includes a regression fit line with mean fitted values and related shaded 95% confidence interval shown (n = 103 individuals; 64 NC and 39 SCZ). C80 is the only one with a significant PRS association consistent with diagnosis direction. Source data are provided as a Source Data file. **b** Gene enrichment analysis results are shown for the C80 component. From the bottom, the first (GWAS), second (MAGMA), and third orange grids (H-MAGMA) show enrichment results for schizophrenia risk genes, other psychiatric illness risk genes, and immune condition risk genes. Enrichment testing results are shown for differentially expressed genes, differentially methylated genes, and loss of function variant intolerant genes in the green grid. The final light-blue grid shows C80 tissue specificity as determined by the tissue scores generated during the SDA process and reflects the relative contribution of component gene networks within each of the sampled regions to the overall component. Adjusted p-values shown are empirical p-values obtained from permutation tests (overrepresentation analysis: one-sided Fisher exact test). **c** Venn diagram shows the intersection between C80 genes and SCZ, MDD, and ADHD GWAS risk genes. Blank regions indicate no common genes. In the case of a single gene result, that gene is listed. **d** Cell-type specificity of C80 component using human (left) and mouse (right) single-cell atlases. Mean-rank Gene Set Test in the *limma R* package[115] was used to obtain the enrichment p-values shown. y-axes show FDR-adjusted p-values after correcting for multiple comparisons across components (N = 69) and cell types (human atlas = 10; mouse atlas = 24). Red dashed lines represent $\alpha_{[FDR]}$ = 0.05. Individual data points are shown using overlaid dot plots. Barplots demonstrate a higher specificity for GABAergic, medium spiny, and dopaminergic neurons. Source data are provided as a Source Data file. ADHD attention deficit hyperactivity disorder, ASC astrocytes, ASD autism spectrum disorder, BD bipolar disorder, CD Crohn's disease, CN Caudate Nucleus, DEGs differentially expressed genes, DLPFC dorsolateral prefrontal cortex, DMGs differentially methylated genes, END endothelial cells, HP hippocampus, exCA pyramidal neurons from the hippocampal CA region, exDG granule neurons from the hippocampal dentate gyrus, exPFC pyramidal neurons from the prefrontal cortex, GABA GABAergic interneurons, LoF loss of function intolerant genes, MDD major depressive disorder, MG microglia, NC Neurotypical controls, NSC neuronal stem cells, OCD obsessive-compulsive disorder, ODC oligodendrocytes, OPC oligodendrocyte precursor cells, PRS polygenic risk score as reported by the third wave (primary) analyses of the Psychiatric Genetics Consortium[2]; PTSD posttraumatic stress disorder, SA suicide attempt, SCZ Patients with schizophrenia, UC ulcerative colitis.

discovery C80 and replication C18 components is reported in Supplementary Data 3.

## Brain functional association analysis

Based on these results in support of C80's role in SCZ, SCZ risk, and dopaminergic function, we computed a PRS stratified for genes within C80 (C80-PRS) to examine the relationships between C80-specific SCZ genetic risk burden and neurochemical and neurofunctional parameters in the living human brain (see Methods for details about PRS computation, p-value thresholds used and neuroimaging association analyses). C80-PRS was positively associated with greater striatal dopamine synthesis capacity as measured by [18F]-FDOPA specific uptake in NC and patients with SCZ (C80-PRS1: Fisher's $z_{r[99.5\% \, CI]}$: $0.33_{[0.01, \, 0.65]}$; p = 0.0037; $p_{[Bonferroni]}$ = 0.037; $R^2$ = 0.10) in the whole-striatum ROI analyzed in our discovery cohorts (Table 1; Fig. 4a). In the same ROI, we also found a significant association for PRS2 (C80-PRS2: Fisher's $z_{r[99.5\% \, CI]}$: $0.34_{[0.03, \, 0.66]}$; p = 0.0024; $p_{[Bonferroni]}$ = 0.024; $R^2$ = 0.11) (Supplementary Fig. 3a). Furthermore, both the C80-PRS1 and C80-PRS2 were significantly and more strongly associated with [18F]-DOPA PET uptake in the associative striatum (C80-PRS1: Fisher's $z_{r[99.5\% \, CI]}$: $0.38_{[0.07, \, 0.70]}$; p = 0.0006; $p_{[Bonferroni]}$ = 0.006; $R^2$ = 0.13 C80-PRS2: Fisher's $z_{r[99.5\% \, CI]}$: $0.35_{[0.03, \, 0.66]}$; p = 0.002; $p_{[Bonferroni]}$ = 0.02; $R^2$ = 0.11), (Supplementary Fig. 3b, c). There was no significant correlation with limbic or sensorimotor striatum when correcting for multiple comparisons. Interestingly, these results remained consistent even across different genetic ancestry definitions (Supplementary Fig. 4a, b; see Supplementary Methods for details).

In our independent replication cohort (Table 1), C80-PRS was also positively associated with greater striatal [18F]-FDOPA specific uptake, albeit at a different PRS threshold (C80-PRS6: $t_{[149]}$ = 3.95; k = 16; $p_{[FWE]}$ < .05; $R^2$ = 0.10). This association was predominantly observed in the sensorimotor region extending into the associative striatum at p < 0.005 (Fig. 4a).

Finally, we investigated the association of C80-PRS with striatal functioning using fMRI in participants who performed a reward processing task (see Methods and Supplementary Methods for details). We found that the C80-PRS1 was positively correlated with differential anticipatory activation in the right associative striatum during high vs low motivation assessed in a discovery sample of 86 NC (Table 1; C80-PRS1: $p_{[TFCE-FDR]}$ = 0.04; Z = 3.54; x = 17; y = 15; z = −5; 60 voxels; partial $R^2$ = 0.13). Specifically, participants with higher C80-PRS, and thus higher predicted striatal dopamine synthesis, showed greater striatal activation when they expected a reward during the task (Fig. 4b). We consistently found this association in an additional independent sample of 55 NC (Table 1; C80-PRS1: $p_{[TFCE-FDR]}$ = 0.03; Z = 3.75; x = 18; y = 17; z = 5; 34 voxels; partial $R^2$ = 0.25) in a cluster located once again in the associative striatum (voxel-wise discovery-replication overlap volume: 162 mm³; see Supplementary Methods and Supplementary Fig. 7). It is worth mentioning that we confirmed the C80-PRS effects also on the BOLD signal extracted from both the right and left-associative striatum ROIs (see Supplementary Methods for details). Importantly, the neuroimaging associations we identified with C80-PRS were not significant for the other components associated with genetic risk for SCZ (see Supplementary Methods and Supplementary Fig. 6d).

We also computed a measure of cumulative SCZ risk burden based on GWAS risk genes not in C80 (C80-PRS-complementary) and did not find any significant association in any of the PET and fMRI samples (p > 0.05; Supplementary Fig. 5a, b). It is also worth mentioning that the number of SNPs included at each threshold for the C80-PRS was always lower than for the C80-PRS-complementary (Table 2), indicating the SNPs mapped to C80 genes represent a minority more closely involved in dopaminergic processes than the rest. Supplementary Data 4 includes SNPs mapping to C80 genes used to compute the C80-PRSs.

## Discussion

The genetic architecture of SCZ is complex and spans the genome[2]. Despite evidence for aggregation of implicated genes into certain clusters[43], characterizing the functional biology of SCZ risk genetics has been a challenge. We applied a tensor decomposition method, i.e., SDA, to post-mortem brain gene expression data from three brain regions, i.e., CN, HP, and DLPFC. We identified cohesive biological pathways that are implicated in SCZ illness and risk. Such pathways delineate plausible routes from SCZ genetic risk variation to specific neural circuit functions perturbed in this condition. We discovered a CN-dominant co-expression gene set (C80) that is enriched for genes differentially co-expressed in individuals with SCZ relative to NC and is associated with individual genetic risk for SCZ, features that suggest a role in SCZ pathogenesis. Expanding long-held hypotheses of dopaminergic involvement in psychosis in general and in SCZ more specifically, this gene set showed enrichment for dopamine-system genes and embedded SCZ risk variation that specifically tracked in vivo neurochemical and neurofunctional dopamine- and illness-related phenotypes.

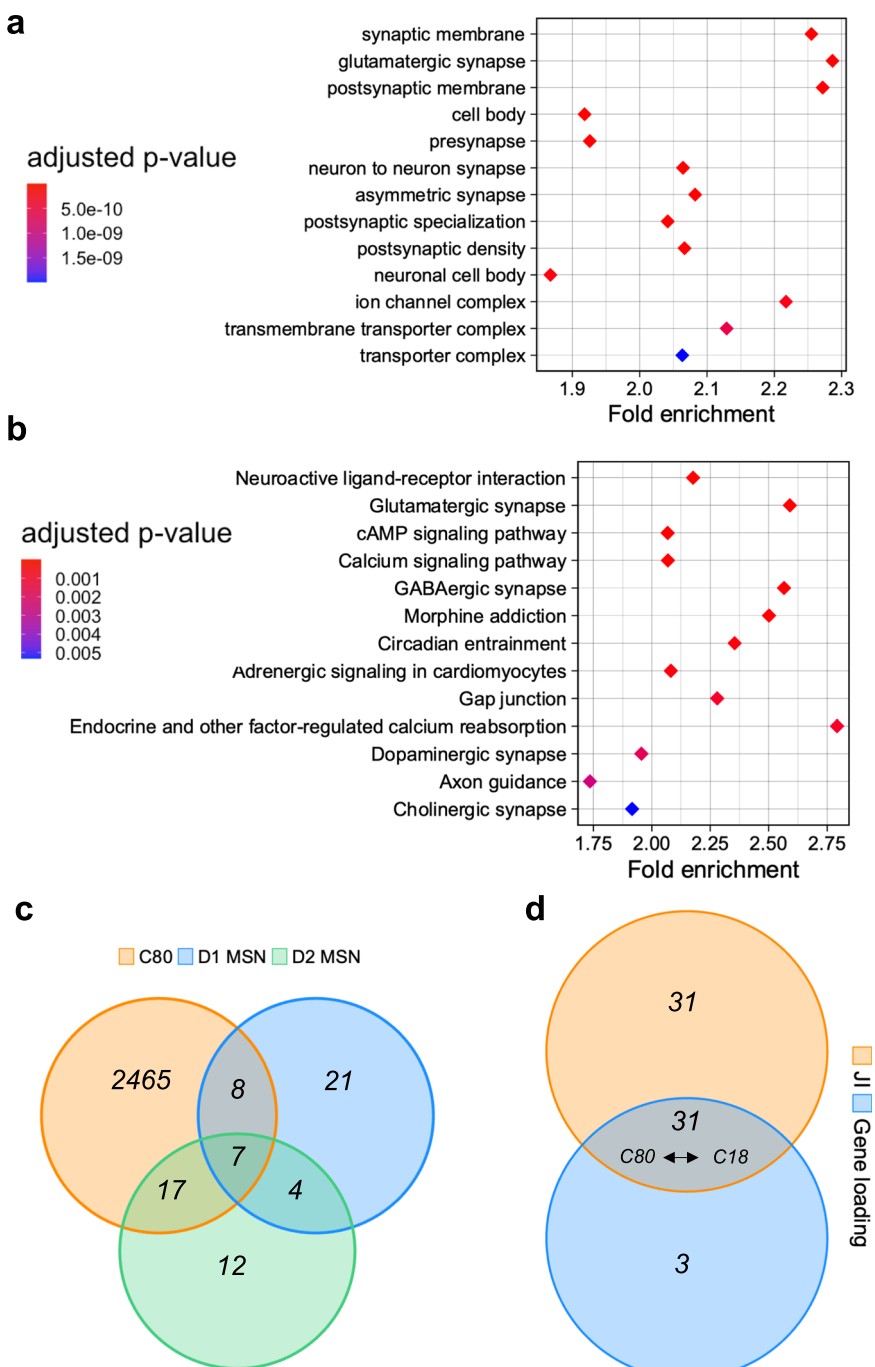

**Fig. 3 | Synaptic dopaminergic specificity of C80. a, b** Gene ontology (cellular compartment) and KEGG enrichment of C80 for both pre and post-synaptic compartments as well as dopaminergic, GABAergic, and glutamatergic synapses. Overrepresentation analysis was performed using the *clusterProfiler R*[116] package and FDR-adjusted *p*-values are reported. Diamonds represent fold enrichment (*x*-axis) for each Gene ontology category (*y*-axis) and are colored based on the respective adjusted *p*-value. **c** Venn Diagram shows the intersection between C80 genes and genes expressed in subpopulations of D1- and D2-expressing MSNs in the nucleus accumbens as defined by Tran, Maynard[40]. A larger intersection is found with D2-MSN than D1-MSN. **d** Overlap between SDA components generated from the LIBD and GTEx datasets that are significantly replicated (one-sided permutation test; empirical *p*-value < 0.001) using JI or gene loading correlation. Discovery C80 and replication C18 are one of the 4 pairs of components consistent with both JI and gene loading. JI jaccard index, MSN medium spiny neurons.

The SDA algorithm provided an efficient technique to uncover sparse gene networks that were not only statistically robust but also biologically coherent. The C80 component captured gene expression covariance showing biological specialization for striatal dopaminergic circuitry implicated in SCZ. A key component of this circuitry, the C80 member gene *DRD2*, is expressed both at the presynaptic and post-synaptic terminals. The gene product is the D$_2$ dopamine receptor, acting as an autoreceptor regulating dopamine synthesis and release in

the presynaptic terminal, and driving the indirect striatal pathway activity in the postsynaptic terminal of MSNs[44]. In line with *DRD2*'s autoreceptor role, SDA segregates *DRD2* together with the genes for the primary dopamine biosynthesis enzymes, *TH* and *DDC*, in a single, SCZ-associated component, although isoform transcript analyses did not confirm an isoform preference replicated across LIBD and GTEx data. This is not surprising in principle, as greater dopamine biosynthesis should have downstream effects on both receptor isoforms.

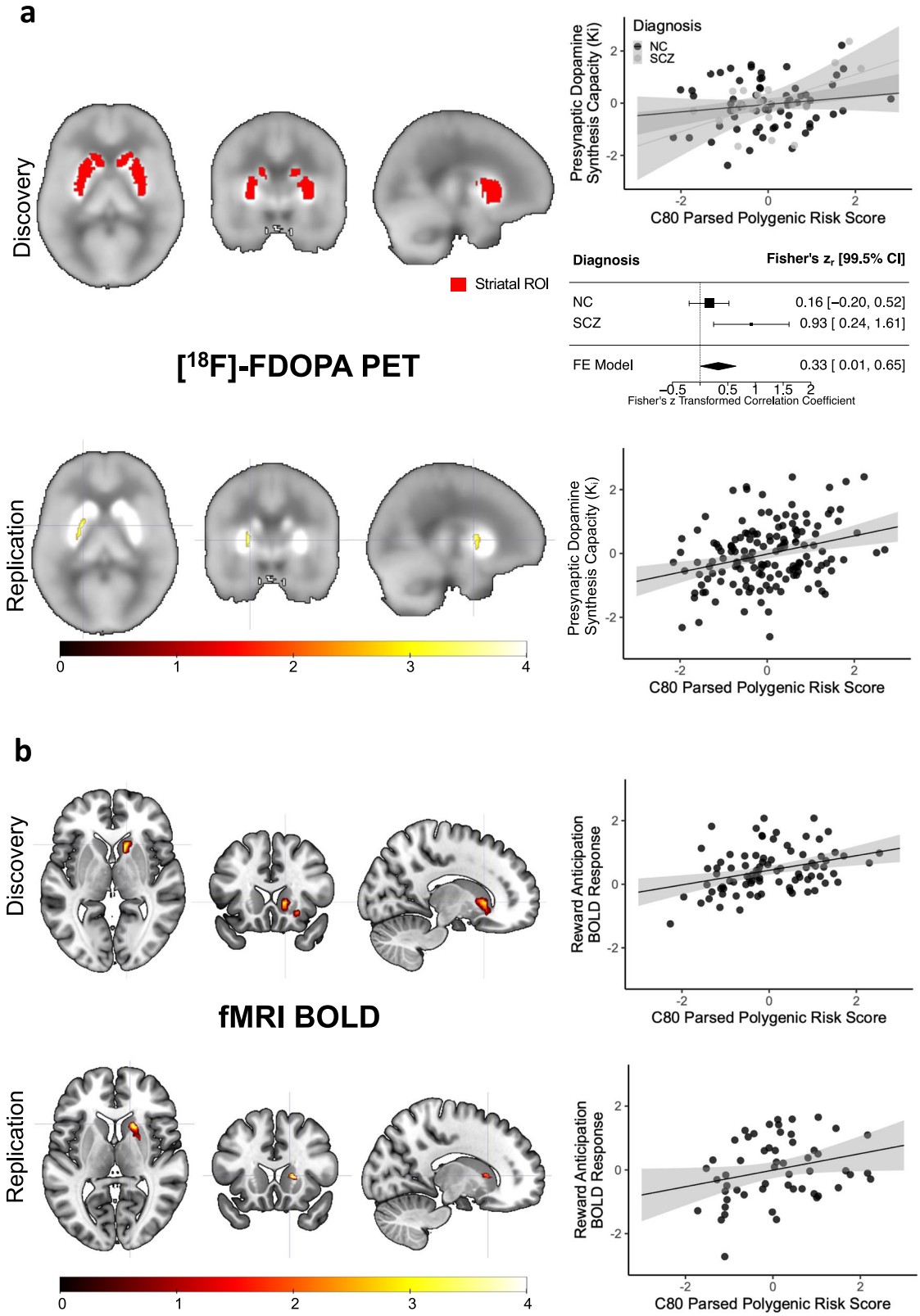

However, at the total gene expression level, greater C80 (higher SCZ risk) correlated with higher striatal expression of *DDC*, translating into an increase of dopamine synthesis, and as expected, lower expression of *DRD2*, with both results replicated in an independent NC cohort. These findings, along with the neurofunctional results, highlight the control of presynaptic dopamine synthesis and release as a mechanism of dopamine-associated pathogenesis[21]. Importantly, while previous

work on translating genetic risk into gene expression association has highlighted the presynaptic short isoform as the molecular mechanism of risk, here we are looking at co-expression in a broader biological context than genetic risk alone at a single locus.

It is notable that follow-up analyses of C80 gene membership additionally identified a preference for genes expressed in the indirect pathway, i.e., D$_2$ dopamine receptor bearing MSNs (e.g., *DRD2*, *PENK*)

**Fig. 4 | C80-PRS association with neuroimaging parameters: striatal dopamine synthesis capacity ([18F]-FDOPA PET) and reward anticipation-related fMRI activation (fMRI BOLD). a** Associations between C80-PRS and both PET cohorts are shown. First row (PET discovery; $n = 84$ individuals; 64 NC and 20 SCZ): on the left whole-striatum region of interest (ROI) coverage (red) is shown overlaid on a grayscale standardized [18F]-FDOPA PET activity map; on the right graphs shows standardized individual mean $K_i$ values for this ROI plotted against C80-PRS for the neurotypical control and SCZ subjects (upper) as well as the forest plot of the metanalysis (lower). Second row (PET replication; $n = 150$ NC): Region of a positive association between C80-PRS and presynaptic dopamine synthesis capacity ([18F]-FDOPA $K_i$) is shown as a statistic parametric map (color indicates $t$-statistic value) overlaid on a grayscale standardized [18F]-FDOPA PET activity map ($p < 0.005$, uncorrected for display). The scatter plot shows standardized individual mean $K_i$ values for a 2 mm sphere around the peak voxel plotted against C80-PRS. Mean fitted values and related shaded 95% confidence interval are shown in the scatterplots. Fisher's r-to-z transformed correlation coefficients and related 99.5% confidence interval are shown in the forest plot. Source data are provided as a Source Data file. **b** Associations between C80-PRS and both fMRI cohorts are shown. First (fMRI discovery; $n = 86$ NC) and second (fMRI replication; $n = 55$ NC) rows: Regions of positive association between C80-PRS and fMRI BOLD response during reward anticipation are shown as statistic parametric maps (color indicates the threshold-free cluster enhancement (TFCE) statistics expressed in the −log10 scale). All results meet thresholds of $p_{[TFCE-FDR]} < 0.05$ and cluster extent >20 voxels. Scatter plots show standardized individual MID-related fMRI BOLD contrasts plotted against C80-PRS with mean fitted values and related shaded 95% confidence interval shown. Source data are provided as a Source Data file.

**Table 2 | Number of SNPs used for parsed PRS**

| Cohort | Diagnosis (NC/SCZ) | PRS1 (5e-08) | | PRS2 (1e-06) | | PRS3 (1e-04) | | PRS4 (.001) | | PRS5 (.01) | | PRS6 (.05) | |
|---|---|---|---|---|---|---|---|---|---|---|---|---|---|
| | | C80 | compl | C80 | compl | C80 | compl | C80 | compl | C80 | compl | C80 | compl |
| [18F]-FDOPA PET Discovery | NC | 89 | 141 | 174 | 277 | 588 | 1234 | 1347 | 3029 | 3444 | 8843 | 7309 | 19,771 |
| | SCZ | 83 | 136 | 157 | 258 | 528 | 1088 | 1188 | 2644 | 2952 | 7604 | 6192 | 16,775 |
| [18F]-FDOPA PET Replication | NC | 82 | 137 | 157 | 272 | 569 | 1186 | 1337 | 3001 | 3624 | 9394 | 8215 | 22,094 |
| Reward fMRI Discovery | NC | 84 | 142 | 172 | 284 | 608 | 1276 | 1458 | 3268 | 3969 | 10,228 | 8982 | 24,090 |
| Reward fMRI Replication | NC | 89 | 144 | 177 | 291 | 647 | 1328 | 1540 | 3464 | 4214 | 10,825 | 9577 | 25,608 |

Number of SCZ risk SNPs used for the C80-PRS and C80-PRS-complementary (compl) for each cohort analyzed in the brain functional association analysis. Here are shown the first six GWAS $p$-value thresholds considered. Source data are provided as a Source Data file.

over those expressed in direct pathway associated, i.e., D₁ dopamine receptor bearing MSNs (e.g., *DRD1*, *PDYN*)[41]. As KEGG pathway analysis revealed a collection of dopaminergic, GABAergic, glutamatergic, and cholinergic pathway-related genes belonging to this component, it is interesting that among the genes segregated by SDA in both C80 and C18 components is the one encoding for the M4 muscarinic receptor (*CHRM4*). This receptor was previously associated with the regulation of cholinergic and dopaminergic neurotransmission in SCZ[45–48] and was recently highlighted as a potential therapeutic target for this disorder[49–51]. Besides the relevance of C80 to presynaptic dopaminergic mechanisms, these observations point to a wider biological interpretation of the genes co-expressed within this SCZ-associated component, including cortico- and nigro-striatal terminals closely tethered to the indirect pathway.

Following the identification of these links between C80 and dopaminergic systems, we conducted in vivo neuroimaging investigations and found that SCZ genetic risk variation that is mapped to the C80 gene set—and not cumulative risk outside of it—is specifically associated with elevated presynaptic dopamine synthesis in the striatum, which we observed in both NC and SCZ cohorts across three independent PET samples totaling 235 participants. This is consistent with the [18F]-FDOPA-associated phenotype in SCZ[52] and further supports the notion that C80, also expressed to a greater degree in SCZ, plays an important role in presynaptic dopamine dynamics. Thus, the present results may provide a molecular mechanism for the risk signature of a central phenotypic pillar of the modern dopamine hypotheses of SCZ[52].

The specificity of these findings sheds light on the elusive source of heterogeneity in SCZ and its pathobiology. These data align with the notion that some routes to clinical illness (e.g., those within the C80 pathway) preferentially perturb presynaptic dopamine systems over others. This model provides a possible molecular basis for the observation that not all individuals with SCZ show excessive presynaptic dopamine synthesis capacity and not all patients respond well to antidopaminergic medications[18,19]. More generally, the approach we employed may be promising for stratifying patients based on their pathway-specific genetic liability to illness, which, if confirmed to be clinically informative, could provide new avenues for personalized medicine.

The association between these pathway-specific variants and heightened striatal dopamine synthesis, as evidenced by post-mortem and PET data, aligns with findings from a dual PET-fMRI study that demonstrated positive correlations between reward anticipation-related activation and striatal dopamine release in healthy individuals[53]. Accordingly, relative dopamine depletion attenuates striatal activation during the same task in healthy subjects[54]. We found convergent positive associations between SCZ risk within the dopamine-system-enriched C80 gene set and anticipatory activation in both our discovery and replication cohorts. The activation clusters localized to the head of the caudate, the same region used in the postmortem study. Moreover, our results are consistent with past findings of positive correlations between polygenic risk for SCZ and striatal activation during the MID task in a large sample of healthy adolescents[55]. Importantly and again, cumulative risk outside of this filter (i.e., variants not included in C80) did not show a significant relationship with this anticipatory BOLD response; along with the similar pattern observed in our PET results, this specificity suggests that parsing the PRS into co-expression pathways can provide a biologically accurate physiological modeling to translate genetic risk into brain mechanisms[43].

The positive correlations between C80-specific SCZ risk burden and reward anticipation BOLD response in neurotypical individuals deviate in direction from prior findings of blunted striatal response to reward anticipation in patients with SCZ[24,25,56–60]. The difference between genetic and clinical findings may have multiple sources, including illness characteristics and pharmacological treatment. Patients with SCZ display abnormal salience attribution patterns[61], which could lead to reduced contrast between anticipation cues and

baseline, ultimately resulting in poorer motivational performance and BOLD contrast during reward anticipation[62–64]. Secondly, the activity of different brain regions involved in the reward system can be affected by the disorder[65], while being preserved in our fMRI samples only including NC. Third, SCZ patients exhibit elevated striatal dopamine synthesis and release as measured by PET[7,11–14], suggesting that higher steady-state dopamine levels may cause an apparent blunted response by elevating baseline activation. Consistent with this hypothesis, Knutson et al.[66] reported that amphetamine administration, which blunts task-based dopamine release while enhancing steady-state availability in the striatum[67], leads to decreased peak activation but prolonged activation duration during reward anticipation in healthy individuals. Taken together, we suggest that the blunted response to stimuli in a saliency-modulating task observed in SCZ may arise at least in part from both reward devaluation as well as enhanced steady-state baseline activity.

A further paramount consideration in studies of reward circuit in patients with SCZ is the impact of neuroleptic medications on the sensitivity of the brain's reward system[68,69]. Previous studies indicate that antipsychotic drugs can blunt reward-related anticipatory striatal activation in individuals with SCZ[56,57]. This effect may be associated with the blockade of $D_2$ dopamine receptors[56,57] or the suppression of dopaminergic neuron firing[70–72], known as inactivation block. In fact, the recent report by Benjamin et al.[21] highlights a significant downregulation of the *TH* gene in CN of SCZ patients receiving antipsychotic drugs. Acute depletion of dopamine has been associated with reduced striatal activation during reward anticipation in both patients[73] and controls[54]. Notably, atypical antipsychotics, which exhibit a lower level of $D_2$ receptor affinity, were found to enhance striatal fMRI BOLD signal during reward anticipation relative to first-generation, high-affinity $D_2$ blocking agents[57,74,75]. Nonetheless, even atypical antipsychotic medications may increase baseline striatal activity in a dopamine-dependent fashion[76]. Thus, the effects of illness and pharmacological stimulation are not necessarily aligned with the relationship between illness genetic risk and striatal physiological activation in the neurotypical state. In this regard, examining the effects of polygenic risk for SCZ in samples of healthy controls provided an important perspective on risk biology while avoiding important illness-associated confounders, such as treatment with $D_2$ dopamine receptor antagonists.

Limitations of this study include the relatively small sample size used in the gene co-expression analysis, which is pivotal for decomposition approaches[37]. To obtain the 3D tensor used as input for SDA, we had to exclude samples without available data in all three brain regions analyzed. This filter was especially limiting for the GTEx replication dataset. It is also important to acknowledge unavoidable dataset discrepancies that might have hindered replications in the *DRD2* transcript-level analysis. Table 1 delineates the demographic heterogeneity of LIBD samples in contrast with the more homogenous GTEx cohort, especially in age distributions (GTEx: $59.2 \pm 9.6$ years; LIBD: $46.2 \pm 16.9$ years; Wilcoxon rank sum test: one-tailed $p < 2.2 \times 10^{-16}$). Additionally, the substantial difference in sequencing depth (LIBD: mean of 125.2 million reads per sample[21,77]; GTEx: mean of 50 million[42]) suggests a different resolution across the two datasets in detecting fine-grained transcript-level variations. Despite these distinctions in demographic and technical aspects, at the total gene level, our tensor decomposition analysis successfully identified consistent co-expression patterns related to dopaminergic neurotransmission across both datasets. These results suggest that the finer biological resolution required for *DRD2* isoform association might be more sensitive to the variations inherent in the datasets. Moreover, while including SCZ post-mortem data in our analyses provided important insights into illness associations of identified SDA components, antipsychotic treatment, and other illness-associated epiphenomena may have introduced confounding effects for the SDA analysis. We tried to address this issue by performing SDA on the same three tissues using

an independent dataset consisting of only healthy controls (GTEx) to assess the rate of replication and generalizability of these results. Indeed, we found that 90% of the components identified were replicated, and the one we studied was very well reflected in GTEx. Furthermore, while the total sample size for [18F]-FDOPA PET genetic studies in this work is unparalleled, the within-cohort sample sizes are limited, which may explain minor differences in peak findings across different PRS SNP *p*-value thresholds or anatomically within the striatum. Nonetheless, the convergent positive association identified between C80 and striatal presynaptic dopamine synthesis in all cohorts studied despite independent, multi-site sampling and diverse methods bolsters confidence in the PET results. Additionally, while the non-invasive reference region graphical linearization method used here for [18F]-FDOPA quantification has been well validated, it is possible that alternative kinetic modeling in future studies using arterial data may allow for a more comprehensive view of observed effects. Finally, it is possible that sample size limitations in the fMRI dataset coupled with the substantial variability in the degree and direction of laterality across individuals might have affected the localization of the effects and prevented the identification of weaker but important bilateral activations at the voxel-wise level, although the consistency of findings across independent cohorts and at the ROI level mitigates this concern.

In conclusion, these results highlight a dopamine-related striatal gene set that characterizes the illness state in SCZ, is implicated in SCZ genetic risk, and is involved in dopamine synthesis and striatal physiological activity in vivo, suggesting that genetic risk within this pathway differentially affects SCZ-relevant striatal function. These observations provide evidence that polygenic risk for SCZ can be effectively parsed into pathways important for specific systems-level functions that are measurable even in the absence of illness[43]. Furthermore, they suggest a molecular basis of how genetic risk within the C80 pathway might affect illness-relevant striatal neurochemistry and neurofunction and may open new possible avenues for studying clinical heterogeneity[18,43] and drug treatment response[43].

## Methods

### Ethics

The research described herein complies with all relevant ethical regulations. Postmortem human brain tissue was obtained as previously described[77,78]. Briefly, tissues were primarily obtained by autopsy from the Offices of the Chief Medical Examiner of the District of Columbia and of the Commonwealth of Virginia, Northern District, all with informed consent from the legal next of kin (protocol 90-M-0142 approved by the National Institute of Mental Health (NIMH)/National Institutes of Health (NIH) Institutional Review Board). The National Institute of Child Health and Human Development Brain and Tissue Bank for Developmental Disorders (https://medschool.umaryland.edu/BTBank) provided infant, child, and adolescent brain tissue samples under the NO1-HD-43368 and NO1-HD-4-3383 contracts. Additionally, donations of postmortem human brain tissue from patients with SCZ were provided with informed consent by next of kin from the Office of the Chief Medical Examiner for the State of Maryland under protocol number 12–24 from the State of Maryland Department of Health and Mental Hygiene and from the Office of the Medical Examiner, Department of Pathology, Homer Stryker, Maryland School of Medicine under protocol number 20111080 from the Western Institute Review Board. The Institutional Review Board of the University of Maryland at Baltimore and the State of Maryland approved the study protocol. The Lieber Institute for Brain Development (LIBD) received the tissues by donation under the terms of a material transfer agreement.

The discovery cohort of participants in the PET study was obtained under ethical permission given by the Administration of Radioactive Substances Advisory Committee (ARSAC), the South

London and Maudsley/Institute of Psychiatry NHS Trust, the London Bentham Research Ethics Committee, and the Hammersmith Research Ethics Committee. All participants provided written, informed consent per King's College London (KCL) IRB-approved protocols.

The replication cohort of participants in the PET study was obtained under ethical permission given by the National Institute of Mental Health Institutional Review Board and the National Institutes of Health (NIH) Radiation Safety Committee. All participants provided written, informed consent per NIH IRB-approved protocols.

The discovery cohort of participants in the fMRI experiments had no history of any psychiatric or neurological disorders and gave written, informed consent for a protocol approved by the NIH Combined Neurosciences IRB. Participants were told that they would be monetarily compensated based on earnings in the task.

The replication cohort of participants in the fMRI experiments had no history of any psychiatric or neurological disorders and gave informed consent for a protocol approved by the institutional ethics committee of the University of Bari Aldo Moro (UNIBA). Participants were told that they would be compensated with one gift gadget (pen, t-shirt, pin, bag, pouch, notebook) when they earned at least 1700 points, and the chance of choosing between two or three gifts of their choice (when reaching 1900 and 2300 points, respectively) and encouraged to respond as quickly as possible.

The study design and conduct complied with all relevant regulations regarding the use of human study participants and was conducted in accordance with the criteria set by the Declaration of Helsinki.

### Lieber Institute for Brain Development (LIBD) post-mortem data−discovery cohort

We used post-mortem human brain tissue from the LIBD Human Brain Repository. Patients with SCZ were collected from the Office of the Chief Medical Examiner for the State of Maryland under the State of Maryland Department of Health and Mental Hygiene Protocol 12−24 and from the Kalamazoo County Michigan Medical Examiners' Office under Western Institutional Review Board Protocol 20111080. All included NC subjects had minimal age-associated neuropathology (determined from postmortem histopathological examination) and no substance or drug use from toxicology and were free from any psychiatric or neurological disorder from clinical histories. Postmortem clinical information was gathered by conducting family interviews with the next of kin. After psychiatric record reviews and postmortem family interviews were completed, brief psychiatric narratives were prepared on each case, summarizing the demographic, clinical, medical, and death information obtained from as many sources as possible (i.e., multiple psychiatric records, police reports, neuropathology reports, medical examiner's information, toxicology screen, and postmortem family interview). Each case was then independently reviewed by two board-certified psychiatrists, who arrived at consensus DSM-IV Axis I lifetime diagnoses or consulted with a third reviewer to reach a final diagnosis. All samples were collected and processed using a standardized protocol specifically developed to minimize sample heterogeneity and technical artifacts[77,78].

The CN samples were derived from the anterior 'head' portion, a subregion tightly connected with the prefrontal cortex; HP samples from the mid-hippocampus proper (all dissections included the dentate gyrus, CA3, CA2, and CA1) plus the subicular complex; and DLPFC samples from Brodmann Area 9/46 at the level of the rostrum of corpus callosum[79].

For all tissues, RNA sequencing was performed via the Illumina Ribozero Kit as previously described[77]. Briefly, RNA was extracted using the QIAGEN AllPrep DNA/RNA Mini kit, which concurrently extracted DNA and total RNA. Following RNA extraction, sequencing libraries were prepared from 300 ng of total RNA using the TruSeq Stranded Total RNA Library Preparation kit with RiboZero Gold rRNA depletion. For quality control, synthetic External RNA Controls

Consortium (ERCC) RNA Mix 1 was spiked into each sample. These paired-end, strand-specific libraries were sequenced on an Illumina HiSeq 3000 at the LIBD Sequencing Facility across multiple lanes. FASTQ files were generated using the Illumina Real-Time Analysis module by performing image analysis, base calling, and the BCL Converter (CASAVA v1.8.2). The reads were aligned to the hg38/GRCh38 human genome (GENCODE release 25, GRCh38.p7, chromosome only) using HISAT2 (v2.0.4)[48] and Salmon (v0.7.2)[49] using the reference transcriptome to initially guide alignment based on annotated transcripts. The synthetic ERCC transcripts were quantified with Kallisto (v0.43.0)[50]. Counts were generated as previously described[77,78]. Briefly, sorted BAM files from HISAT2 alignments were generated and indexed using SAMtools (v1.6; HTSlib v1.6). Alignment quality was assessed using RSeQC (v2.6.4)[51]. Gene-level mRNA expression was quantified as Reads Per Kilobase per Million mapped reads (RPKMs) and annotated as total gene expression separately for each brain region using GENCODE release 25, GRCh38.p7.

We included NC and SCZ samples with European or African American ancestry, all with RNA Integrity Number (RIN) ≥ 6. We used inter-array distance to identify tissue-specific outlier subjects deviating more than three standard deviations from the mean[32] (CN = 4; HP = 7; DLPFC = 5). We then focused our analyses on mRNA expression measurements that were available for common samples ($N = 238$) and genes ($N = 58,037$) across all three tissues. The sex of participants was determined based on self-report and used as a covariate in the following analyses. The demographic data are summarized in Table 1.

### The Genotype-Tissue Expression (GTEx) post-mortem data−replication cohort

We used the *recount3 R*[80] package to download already processed GTEx v8 RPKMs for CN, HP, and DLPFC (Frontal Cortex BA9). Data available for all three tissues consisted of 120 samples and 54,892 genes (Table 1). Sex of participants was determined based on self-report and used as a covariate in the following analyses.

### King's College London (KCL) PET data−discovery cohorts

Two cohorts, one with 92 NC and one with 47 individuals with SCZ, (see Supplementary Table 1 for demographics) underwent [18F]-FDOPA positron emission tomography (PET) scans to measure dopamine synthesis capacity (indexed as the influx rate constant $K_i$) in the striatum as previously described[81–86]. In short, after pretreatment one hour before the scan with fixed doses of carbidopa (150 mg) and entacapone (400 mg) to reduce peripheral tracer metabolism, and immediately following intravenous injection of [18F]-FDOPA, a series of dynamically binned emission frames were acquired over 95 minutes. Computed tomography (CT) imaging was performed for attenuation correction. Scans were obtained on one of the following PET scanners: an ECAT HR + 962 (CTI/Siemens, Knoxville, Tennessee), and an ECAT HR + 966 (CTI/Siemens, Knoxville, Tennessee), and two Siemens Biograph HiRez XVI PET-CT scanners (Siemens Healthcare, Erlangen, Germany). Reconstructed, attenuation-corrected emission scans were realigned to correct for interframe head motion. An atlas defining the regions of interest (striatum, its subdivisions, and the reference region (cerebellum) as described in Howes et al.[15] was co-registered to a tracer-specific template and transformed to each subject's PET data series using SPM 12 software (UCL, London, UK). Time-activity curves were extracted for the regions of interest and entered into standard Patlak-Gjedde graphical linear models using the reference region to adjust for non-specific uptake to obtain the influx rate constant $K_i$, a measure of specific tracer uptake[87]. The primary analyses focused on the whole striatum. For post-hoc exploratory analyses, the striatum was subdivided into limbic, associative (AST), and sensorimotor (SMST) subdivisions based on the functional topography of the striatum and its connectivity as previously described[88]. In short, measurements of the AST were derived as the spatially weighted average of the

precommissural dorsal caudate, precommissural dorsal putamen, and post-commissural caudate. In addition, at the most detailed anatomic level, a significant regional overlap exists between the associative and limbic circuits. Thus, the classification used here identifies these functional circuits only in a probabilistic sense: these regions correspond mostly, but not exclusively, to the various functional subdivisions of the striatum.

The sex of participants was determined based on self-report and used as a covariate in the following analyses.

### National Institute of Mental Health (NIMH) PET data− replication cohort

[18F]-FDOPA PET scans were acquired for a total of 150 healthy subjects (demographics in Table 1) as previously described[89]. In short, after a required 6-hour fast to prevent competition for tracer transport to the brain, a 4-hour caffeine/nicotine restriction, and pretreatment with fixed doses of carbidopa (200 mg) to reduce peripheral tracer metabolism, and immediately following intravenous injection of [18F]-FDOPA, a series of dynamically binned emission frames were acquired over 90 minutes. A transmission scan was performed in the same session for attenuation correction. All scans were obtained on a GE Advance PET tomograph operating in 3D mode with a thermoplastic mask applied to help restrict head movement. Reconstructed, attenuation-corrected emission scans were realigned to correct for interframe head motion. Spatial warping of PET data was performed with ANTs (v2.5.0) software to an MNI space tracer-specific template. A 10 mm Gaussian kernel smoothing was applied to improve voxel-wise signal-to-noise ratios. Using PMOD (v4.4) software (https://www.pmod.com/), time-activity curves from voxels within the striatum were subjected to standard Patlak-Gjedde graphical linear modeling using cerebellar reference region time-activity data as an input function to yield $K_i$ as above[87].

The sex of participants was determined based on self-report and used as a covariate in the following analyses.

### LIBD fMRI data−discovery cohort

An independent sample of 86 NC (demographics in Table 1) participated in an fMRI experiment in which participants performed a modified version of the Monetary Incentive Delay (MID) task[90] based on the expectancy theory of motivation[91]. Participants had no history of any psychiatric or neurological disorders and gave written, informed consent for a protocol approved by the NIH Combined Neurosciences IRB. Participants were told that they would be monetarily compensated based on earnings in the task. Details about the task layout are reported in the Supplementary Methods.

fMRI scans were acquired through a 3 T GE Signa scanner. Gradient-recall echo-planar imaging was used with the following parameters: TR = 2000 ms; TE = 28 ms; flip angle = 90; 64 × 64 matrix; FOV = 240 mm; and 35 3.5 mm slices acquired with an interleaved order of slice acquisition and first five frames discarded to allow steady-state magnetization. Slice timing correction, six-parameter co-registration to adjust for movement, mean functional-image driven spatial normalization to MNI space, and spatial smoothing with an 8 mm Gaussian kernel were applied and yielded time-series data with 3 mm isotropic resolution through SPM12 as implemented in MATLAB (v2023a; https://www.mathworks.com/products/matlab.html). A separate general linear model (GLM) was specified for each participant, modeling time-locked BOLD responses to high reward vs low reward, i.e., low expectation vs. high expectation of reward event onsets, high motivation vs low motivation, by convolving the onset vectors with a synthetic hemodynamic response function as implemented by SPM12. At the model estimation stage, the data were high-pass filtered with a cutoff of 128 s to remove low-frequency drifts. Global scaling was not applied to the data.

The sex of participants was determined based on self-report and used as a covariate in the following analyses.

### UNIBA fMRI data−replication cohort

A cohort of 55 NC (demographics in Table 1) participated in an fMRI experiment in which participants performed an alternative version of the MID task[90] to the one described before. For details about the task layout see Supplementary Methods.

fMRI scans were acquired through a 3 T Philips Ingenia scanner. Gradient-recall echo-planar imaging was used with the following parameters: TR = 2000 ms; TE = 38 ms; flip angle = 90; 64 × 64 matrices; FOV = 240 mm; and 38 3.6 mm slices acquired with an interleaved order of slice acquisition. Slice timing correction, six-parameter co-registration to adjust for movement, mean functional-image driven spatial normalization to MNI space, and spatial smoothing with a 9 mm Gaussian kernel were applied and yielded time-series data with 3 mm isotropic resolution through SPM12 as implemented in MATLAB (v2023a; https://www.mathworks.com/products/matlab.html). A separate GLM was specified for each participant, modeling time-locked BOLD responses to high reward vs low reward event onsets, i.e., high motivation vs low motivation, by convolving the onset vectors with a synthetic hemodynamic response function as implemented by SPM12. At the model estimation stage, the data were high-pass filtered with a cutoff of 128 s to remove low-frequency drifts. Global scaling was not applied to the data.

The sex of participants was determined based on self-report and used as a covariate in the following analyses.

### Genotype data processing and polygenic risk score (PRS) calculation

Genotype data acquisition, imputation, and processing as well as calculation of genomic eigenvariates (GEs) for population stratification were performed in each cohort separately. See Supplementary Methods for further details.

We indexed the whole-genome genetic risk for SCZ by computing the PRS for each sample using the PRSice-2 (v2.3.3) software[92]. To obtain a highly informative SNP set with as little statistical noise as possible, we excluded uncommon SNPs (MAF < 1%), low-quality variants (imputation INFO < 0.9), indels, and SNPs in the extended MHC region (chr6:25−34 Mb). We used PGC (wave 3; primary autosomal analysis) GWAS[2] summary statistics that did not include any of the LIBD discovery samples (leave-sample-out) to weight SNPs by the effect size of association with SCZ. We used European samples from the 1000 Genomes Project[93] (1000 G) as an external reference panel to improve the linkage disequilibrium (LD) estimation for clumping. Both PGC3 leave-LIBD-out and the reference panel were in reference to human genome Build 37.

To stratify SCZ genetic risk for genes within a specific component we first mapped European 1000 G SNPs at 100kbp up- and downstream of each component-specific gene using the MAGMA tool (v1.09b), we then matched component-specific SNPs with PGC3 leave-LIBD-out summary statistics and finally computed the scores for the KCL, NIMH, LIBD- and UNIBA-fMRI cohort separately using PRSset[94] and again the European 1000 G as LD reference panel. As negative control, we computed complementary scores including all PGC3 leave-LIBD-out SNPs not mapping to any of the component-specific genes.

We used PRSs based on 10 SNP sets corresponding to GWAS SNP association $p$-values of $p = 5 \times 10^{-8}$ (PRS1), $p = 1 \times 10^{-6}$ (PRS2), $p = 1 \times 10^{-4}$ (PRS3), $p = 0.001$ (PRS4), $p = 0.01$ (PRS5), $p = 0.05$ (PRS6), $p = 0.1$ (PRS7), $p = 0.2$ (PRS8), $p = 0.5$ (PRS9), and $p = 1$ (PRS10)[2]. Table 2 shows the number of SNPs used for each cohort for each PRS threshold.

### RNA data processing

To analyze LIBD postmortem data with SDA, we first removed mitochondrial genes and genes with RPKM expression median lower than 0.1 or deviating more than 3 standard deviations from the mean in each tissue. We then removed genes with more than 20% zeroes in all three tissues as previously done[35]. We log-transformed RPKM values with an offset of 1, i.e., log2(RPKM + 1). After performing quantile

normalization to normalize samples based on their gene expression, we rank-normalized gene expression using Blom formula[27,28,95] to limit the impact of deviations from normality in expression data. We performed all normalization steps for each tissue separately. The final tensor of 22,356 mRNA expression measurements in 238 samples and across CN, HP, and DLPFC was used as input for SDA (see Table 1).

## Sparse decomposition of arrays (SDA) computation

The SDA algorithm is developed in a Bayesian framework and uses a sparse 'spike and slab'[96] prior to allow the gene loadings of each component to have a unique level of sparsity. This allows us to shrink gene effects to zero so that we can infer more clearly which genes are involved in gene networks. We iterated the algorithm 10 times, and for each run, we obtained latent components defined by three bidimensional matrices: (i) the individual score matrix, which represented the magnitude of the effect of each component across individuals and was used to compute the association with diagnosis; (ii) the tissue score matrix, which indicated the activity of the component for each tissue and was used to identify the contribution of each tissue to the components; (iii) the gene loading matrix, which indicated the contribution of each gene to components and served to identify genes specific to tissues or shared between them (see Supplementary Methods for further details about parameters used).

We thus obtained robust components found consistently across multiple iterations, whereas others only occur in one or a few of them. We clustered similar components across different iterations following published procedures[35] (see Supplementary Methods). We obtained 126 large clusters containing components from multiple different iterations and combined components within each cluster by taking the mean of the individual scores, tissue scores, and gene loadings. We finally used the resulting 126 combined clusters as the basis for further analyses.

## Tissue activity evaluation

We evaluated the tissue loadings across components by column-wise scaling the tissue score matrix (components as columns and tissues as rows) obtained by the SDA decomposition so that the largest score was equal to 1 and the lowest to −1 using a threshold of |0.5| (as previously reported by SDA developers[35]) to infer the tissue specificity of each component and how many components are shared across tissues (Fig. 2b and Supplementary Fig. 1a).

## Confounder analysis

Since SDA identifies non-sparse components that might be expected to arise from confounding effects, we expected singular value decomposition to reveal latent confounders. Singular value decomposition in its principal component analysis implementation has been used often for this aim[35,97–99]. To identify components most likely to represent confounding effects, we computed a series of multiple linear regressions using as dependent variable individual scores from the individual score matrix for each of the 126 SDA components and as predictors both biological confounders (age, sex, diagnosis, first 10 GEs) and technical confounders (postmortem interval (PMI), RIN, mitochondrial mapping rate, rRNA rate, gene mapping rate).

We adopted a confounder detection approach consistent with the reference paper[35] by using the same confounder effect size of 0.274 ($p = 10^{-10}$; sample size = 845). We found that the same effect size corresponded to $p < 5 \times 10^{-4}$ in our sample of 238 individuals (observed power = 80%) and removed the 57 components associated at this threshold with at least one of the technical confounders or GEs. We focused our further analyses on the remaining 69 components.

## Diagnosis and PRS association

To investigate whether the 69 components were differentially co-expressed between NC and SCZ, we tested the association of the component-specific individual scores with diagnosis via ANCOVA while covarying for biological (age, sex, and ancestry) and tissue-specific technical confounders (PMI, RIN, mitochondrial mapping rate, rRNA rate, gene mapping rate), taking into account the component tissue activity. Samples with age >17 were included ($n = 229$) as this was the minimum age in the SCZ sample.

We further evaluated the association of the differentially co-expressed components with PRS via multiple linear regression again covarying for age, sex, diagnosis, and tissue-specific confounders and including only samples with European ancestry ($n = 103$) since the summary statistics used are mainly based on the European population. For this analysis one-tailed tests were used because of the constraint on effect directionality, i.e., we discarded potentially significant results in the opposite direction of diagnosis association. We focused on PGC3 variants with a SCZ association $p$-value < 0.05, since this PRS has been shown to have the highest prediction accuracy for diagnostic status in multiple independent samples[2]. We used Benjamini-Hochberg false discovery rate (FDR) correction to correct for multiple comparisons across SDA components and set $\alpha_{FDR} < 0.05$.

## Biological and functional enrichment analysis

We explored the functional and biological significance of these components through enrichment analyses for multiple psychiatric disorders and immune disorders' top risk loci genes, i.e., putative causal genes identified by setting a fixed distance around each index GWAS-significant SNP and subsequently integrating genomic functions or chromatin interactions[100,101] (ADHD−attention deficit hyperactivity disorder[102]; ASD−autism spectrum disorder[102]; BD−bipolar disorder[102]; MDD−major depressive disorder[103]; OCD−obsessive-compulsive disorder[103]; OCD−obsessive-compulsive disorder[103]; SA−suicide attempt[104]; SCZ−schizophrenia[2]; CD and UC−Crohn's disease and ulcerative colitis[105]).

We also computed enrichment for differentially expressed genes (DEGs) obtained from CN[21], HP, and DLPFC[77]; genes proximal to differentially methylated CpG islands (DMGs) in PFC and blood[78,106–110] and loss of function intolerant genes[111]. For DEGs, we performed a brain region-specific enrichment using the appropriate gene list of each tissue. Moreover, for DLPFC DEGs and DMGs enrichment, we computed a meta-analysis of the papers from which we retrieved target genes to obtain module-wise enrichment $p$-values (sum-log Fisher's method). Considering the overlap between the SDA components obtained, we computed permutation statistics to control for multiple comparisons by first creating for each component a null distribution of 10,000 sets of randomly sampled genes using the 22,356 genes as the universe and then comparing the enrichment hits to the null distribution created from the permuted components (overrepresentation analysis: one-sided Fisher exact test; $\alpha = 0.05$).

## MAGMA analysis

We used the MAGMA tool v1.09b, pathology-specific summary statistics as SNP $p$-value data, and 1000 G European as the reference data file for a European ancestry population to estimate LD between SNPs. We took the following steps: (i) we mapped 1000 G SNPs to genes encompassed in each component (a window of 100 kb upstream and downstream of each gene; for H-MAGMA we used Adult brain Hi-C annotation files already computed in the H-MAGMA publication[39]), (ii) we calculated gene-wide association statistics based on summary statistic SNPs $p$-values (MAGMA "mean" method), (iii) we performed "competitive" gene-set enrichment analysis where the association statistic for genes in the components is compared to those of all other genes with at least one SNP mapped (universe used consisted of 22,356 genes used for SDA). FDR correction served to control for multiple comparisons ($\alpha_{[FDR]} = 0.05$).

## Cell-type specificity analysis

We further asked whether SDA components mapped onto specific brain cell types. We used marker genes already identified by Skene et al. including cell-type specificity indices[112]. They computed specificity indices for each gene ranging between 1 (high specificity for a given cell type) and 0 (low specificity). We used specificity indices derived from single-nuclei RNA-seq of human brains[113], which discriminated ten different cell types (neuron and glia); and from single-cell RNA-seq of mouse brains[112] which encompasses 24 different cell types. We used the Mean-rank Gene Set Test in the *limma (v3.46) R* package[114,115] to evaluate the enrichment of our components for the specificity indices of each cell type. This algorithm performs a competitive test comparing the specificity index rank of the co-expressed genes with the remaining genes. FDR correction served to control for multiple comparisons across components ($n = 69$) and cell types (human atlas = 10; mouse atlas = 24) tested ($\alpha_{[FDR]} = 0.05$).

## Gene ontology analysis

Finally, we explored the gene ontology of components via enrichment analysis through the *clusterProfiler (v3.18) R*[116] package using the Gene Ontology Database (PANTHER, http://pantherdb.org)[117] and the Kyoto Encyclopedia of Genes and Genomes (KEGG, https://www.genome.jp/kegg/) database and setting the 22,356 genes gave as input to SDA as the background set (overrepresentation analysis: one-sided Fisher exact test). FDR correction was applied to control for multiple comparisons ($\alpha_{[FDR]} = 0.05$).

## Association with DRD2, DDC, and TH total gene expression

We evaluated the C80 association with *DRD2*, *DDC*, and *TH* gene expression via a multiple linear regression analysis using C80 individual scores as the dependent variable and *DRD2*, *DDC*, and *TH* expression across CN, DLPFC, and HP as independent variables. We also included age, sex, diagnosis, postmortem interval (PMI), tissue-specific RIN, mitochondrial mapping rate, rRNA rate, and gene mapping rate as covariates. Finally, we added to this model the interaction between diagnosis and each gene expression as well as the interaction between diagnosis and age to control for spurious association driven by postnatal samples (Eq. 1). The gene expression values used were the ones given as input to SDA for all 238 samples (quantile and rank-normalized matrices). We also performed this analysis using only healthy individuals ($N = 154$; Eq. 2).

$$
\begin{aligned}
C80(Y) = &\ Dx * DRD2\,CN + Dx\text{*}DRD2\,DLPFC + Dx * DRD2\,HP \\
&+ Dx * DDC\,CN + Dx * DDC\,DLPFC + Dx * DDC\,HP \\
&+ Dx * TH\,CN + Dx * TH\,DLPFC + Dx * TH\,HP + Dx * Age \\
&+ Race + Sex + PMI + RIN\,CN + RIN\,DLPFC + RIN\,HP \\
&+ TotalAssignedGene\,CN + TotalAssignedGene\,DLPFC \\
&+ TotalAssignedGene\,HP + rRNArate\,CN + rRNArate\,DLPFC \\
&+ rRNArate\,HP + MitoRate\,CN + MitoRate\,DLPFC + MitoRate\,HP
\end{aligned}
\tag{1}
$$

$$
\begin{aligned}
C80(Y) = &\ DRD2\,CN + DRD2\,DLPFC + DRD2\,HP + DDC\,CN + DDC\,DLPFC \\
&+ DDC\,HP + TH\,CN + TH\,DLPFC + TH\,HP + Age + Race\,Sex \\
&+ PMI + RIN\,CN + RIN\,DLPFC + RIN\,HP + TotalAssignedGene\,CN \\
&+ TotalAssignedGene\,DLPFC + TotalAssignedGene\,HP \\
&+ rRNArate\,CN + rRNArate\,DLPFC + rRNArate\,HP \\
&+ MitoRate\,CN + MitoRate\,DLPFC + MitoRate\,HP
\end{aligned}
\tag{2}
$$

## DRD2 transcript-level association analysis

To evaluate the contribution of short and long isoforms to the *DRD2* expression variance, we substituted the *DRD2* terms in the previous gene-level models (Eqs. 1 and 2) with the long and short isoform transcript expression values (Eq. 3).

Transcript counts were preprocessed and normalized to transcripts per million (TPM) estimates as previously described[21,77] and were available for 222 out of the 238 samples previously used. After mapping 138,933 transcripts to the 22,356 genes used for previous analyses, we log-transformed TPM values with an offset of 1, i.e., log2(TPM + 1), and kept transcripts with a median higher than zero. We then performed quantile and rank normalization in each tissue separately as previously done. *DRD2* short isoform survived filters for all tissues while the long isoform had a median higher than zero only in CN. This analysis was also performed using only healthy individuals ($N = 143$).

$$
\begin{aligned}
C80(Y) = &\ Dx * DRD2\,short\,CN + Dx * DRD2\,short\,DLPFC + Dx * DRD2\,short\,HP \\
&+ Dx * DRD2\,long\,CN + Dx * DDC\,CN + Dx * DDC\,DLPFC \\
&+ Dx * DDC\,HP + Dx * TH\,CN + Dx * TH\,DLPFC + Dx * TH\,HP \\
&+ Dx * Age + Race + Sex + PMI + RIN\,CN + RIN\,DLPFC + RIN\,HP \\
&+ TotalAssignedGene\,CN + TotalAssignedGene\,DLPFC \\
&+ TotalAssignedGene\,HP + rRNArate\,CN + rRNArate\,DLPFC \\
&+ rRNArate\,HP + MitoRate\,CN + MitoRate\,DLPFC + MitoRate\,HP
\end{aligned}
\tag{3}
$$

## MSN pathways enrichment

We downloaded the 40 most preferentially expressed genes in each MSN class identified by Tran MN et al.[40] in the nucleus accumbens and focused on $D_1\_A$ and $D_2\_A$ clusters as they represented the largest $D_1$-MSN (67%) and $D_2$-MSN (87%) subclasses, respectively. As 11 genes shared expression for both $D_1\_A$ and $D_2\_A$ clusters, we considered the intersection without these genes for a total of 29 genes in each class. We permuted 10,000 gene sets matching both C80 component size and GC content, gene length, and average expression distributions of C80 genes. The universe from which random genes were pooled consisted of the protein-coding genes given as input to SDA for which this info was available (22,282 genes).

We then computed empirical *p*-values by comparing the enrichment hits with each MSN cluster to the null distribution created from the permuted gene sets ($\alpha = 0.05$).

## GTEx replication analysis

To replicate gene co-expression sets obtained with the LIBD data, we applied SDA on CN, HP, and DLPFC GTEx RNA-seq data using the exact same pipeline previously described. The input matrix for SDA is described in Table 1.

Two replication measures were assessed: correlation between LIBD and GTEx component-specific gene loadings and Jaccard Index (JI) as the intersection/union of the LIBD and GTEx component-specific genes. To identify the LIBD-GTEx pair of replicated components we took for each of the LIBD components the GTEx component with the highest replication measure assessed and iteratively discarded that component to have unique LIBD-GTEx pairs. We then permuted the LIBD components 10,000 times and compared the replication measure previously assessed to the null distribution created from the permuted components to obtain a replication empirical *p*-value for each pair identified ($\alpha = 0.05$).

## Replication of total gene and transcript-level expression in GTEx

To replicate results obtained in the discovery sample, we assessed C18 association with *DRD2* total gene expression via a multiple linear regression analysis using C18 individual scores as dependent variable and *DRD2* expression across CN, DLPFC, and HP as independent variables as previously done in the discovery analysis (Eq. 2).

*DRD2* long and short isoform transcript association was performed as previously done in the discovery analysis (Eq. 3).

We downloaded GTEx v8 transcript TPM values from the GTEx portal (https://gtexportal.org/home/datasets) and after mapping 133,788 transcripts to the 20,475 genes used we performed normalization steps as previously done in the discovery analysis.

## Ancestry stratification

As the summary statistics used are mainly based on the European population and PRS association with other ancestry groups might lead to biases, we evaluated the individual ancestry based on the genotype data rather than only considering the self-reported ancestry for all cohorts included for the brain function association analysis. To this purpose, we used a procedure developed by the ENIGMA consortium that consists of performing a PCA on target data merged with the HapMap[118] phase 3 reference dataset (https://enigma.ini.usc.edu/wp-content/uploads/2012/07/ENIGMA2_1KGP_cookbook_v3.pdf). For this analysis we included all samples whose genotype information was available (KCL: 168; NIMH: 169; LIBD: 86; UNIBA: 2178; see Supplementary Table 1). We then computed an individual ancestry score based on the GE obtained from the PCA analysis. We trained a generalized linear model using the *glmnet R* package; we used the first 20 GE obtained as predictors and the ethnicity (European = 1; Others = 0) as a response variable and considered only samples in the reference dataset. Then, we used the trained model to predict the ethnicity of our samples using the first 20 GE. Finally, for each subject we obtained a European ancestry score and considered a threshold of 90% prediction probability to remove individuals with a non-European ancestry (KCL: 59; NIMH: 3; UNIBA: 213). The remaining samples whose genotype and PET/fMRI data were available were used in further analyses (demographics in Table 1).

Since the KCL discovery cohort was the most heterogeneous in terms of ethnicities included we decided to also evaluate different ancestry subdivisions based on the visualization of the first two PCA dimensions, i.e., top axes of variation (see details in Supplementary Methods).

## Parsed-PRS association with PET data

Considering the sample heterogeneity in the KCL discovery cohort in terms of both ethnicity and scanners used and population type at the diagnostic level, we conducted the association analysis separately in the NC and SCZ samples. A multiple linear regression served to associate the C80 stratified PRS (C80-PRS) as well as its complementary score (C80-PRS-complementary) with [$^{18}$F]-FDOPA uptake in the striatum indexed by Ki using age, gender, cannabis use, scanner and first three GEs as covariates. We then combined the effect of the individual studies with a fixed-effect model meta-analysis, as random-effect models require data to be randomly extracted from equivalent populations, an assumption that does not hold for clinical and control cohorts. We converted t-statistics from the regression model into correlation coefficients using the following formula:

$$r = \sqrt{t^2/(t^2 + DF)}$$

where DF is a number of the degrees of freedom for the t-statistic. We finally used Fisher's r-to-z transformed correlation coefficients as outcome measures in the *metafor*[119] *R* package.

We focused our analysis on dopamine synthesis in an ROI encompassing the whole striatum and to obtain a more granular view of the relationship between risk and phenotypes corrected for multiple comparisons using the Bonferroni method ($\alpha = 0.05/10$). We then explored this association also in different striatum subdivisions as well as across different ancestry definitions (see Supplementary Methods).

As in the KCL analyses described above, we performed separate multiple linear regression analyses for C80-PRS and C80-PRS-complementary predictors in the independent NIMH NC cohort, in

each case using the same covariates as described above in the KCL analysis (i.e., age, sex, and the first three GEs; the 'scanner' variable was omitted, as all scans were obtained on a GE Advance PET scanner). Comparisons were conducted voxel-wise across the whole striatum, using SPM software at a height threshold of $p < 0.05$, voxel-wise family-wise error (FWE) corrected for multiple comparisons.

## Parsed-PRS association with fMRI data

Finally, we associated C80-PRS and C80-PRS-complementary to reward anticipation-related fMRI activation in the independent sample of 86 NC from LIBD. We used the data of 55 NC from UNIBA participants to replicate the results.

BOLD responses to events of interest were modeled separately and time-locked to event onset by convolving the onset vectors coinciding with the onset of events (including cues by type, button press, successful/unsuccessful outcomes, and error trials) with a synthetic hemodynamic response function as implemented by SPM12. For all analyses, the primary outcome measure was the contrast in the BOLD signal of rewarded relative to control cue events, which best reflects reward anticipation responses in this task. Participants additionally completed cognitive testing outside of the scanning environment that assessed full-scale intellective quotient (IQ), which was included as a nuisance covariate in analyses.

Age, sex, IQ, and first three GEs were used as covariates, consistently with previous analyses, whereas MID-related BOLD signal (cue-related anticipatory response during reward versus control trials) was the dependent variable. Cue-related individual contrasts of the 86 NC from LIBD were entered into a group-level analysis to identify voxels with a significant effect of C80-PRS and C80-PRS-complementary on reward anticipation through separate multiple regression performed with SPM12. We considered the threshold-free cluster enhancement correction[120,121] $p_{[TFCE-FDR]} < 0.05$ accounting for multiple comparisons as the number of voxels within the task-related activity mask derived by the one-sample t-test on cue-related individual contrasts ($p_{[FWE]} = 0.05$). Next, the cue-related individual contrasts in the 55 NC from UNIBA were associated with C80-PRS, using age, sex, IQ and first three GEs as covariates. We considered significance at $p_{[TFCE-FDR]} < 0.05$ accounting for multiple comparisons as the number of voxels within the task-related activity mask derived by the one-sample t-test on cue-related individual contrasts ($p_{[FWE]} = 0.05$).

## Reporting summary

Further information on research design is available in the Nature Portfolio Reporting Summary linked to this article.

## Data availability

The LIBD post-mortem raw RNA-Seq FASTQ files for CN, DLPFC, and HP are available through the database of Genotypes and Phenotypes (dbGap) and Globus collections (CN: phs003495.v1.p1 [https://www.ncbi.nlm.nih.gov/projects/gap/cgi-bin/study.cgi?study_id=phs003495.v1.p1]; DLPFC: jhpce#bsp2-dlpfc [http://research.libd.org/globus/jhpce_bsp2-dlpfc/index.html]; HP: jhpce#bsp2-hippo [http://research.libd.org/globus/jhpce_bsp2-hippo/index.html]). The LIBD post-mortem raw genotype data are available through dbGap under accession code phs000979.v3.p2 [https://www.ncbi.nlm.nih.gov/projects/gap/cgi-bin/study.cgi?study_id=phs000979.v3.p2]). The LIBD post-mortem processed RNA-seq data and accession codes to raw RNA-Seq FASTQ files and genotypes used in this study are also publicly available at: https://eqtl.brainseq.org/phase2/ and at: https://erwinpaquolalab.libd.org/caudate_eqtl/. The GTEx post-mortem raw RNA-Seq FASTQ files for CN (GTEx tissue name: Brain-Caudate (basal ganglia)), DLPFC (GTEx tissue name: Brain−Frontal Cortex (BA9)) and HP (GTEx tissue name: Brain-Hippocampus) are available through dbGap with accession code phs000424.v8.p2 [https://doi.org/10.1126/science.1262110]. The GTEx

post-mortem processed RNA-seq data used in this study are publicly available at: https://gtexportal.org/home/downloads/adult-gtex/bulk_tissue_expression. The individual-level raw data for the discovery of PET cohorts from KCL supporting the findings of this study are available from The NeurOimaging DatabasE (NODE) institutional repository (Institute of Psychiatry, Psychology & Neuroscience, King's College London) and can be accessed from co-author O.D.H. (oliver.howes@kcl.ac.uk) upon request. The authors will allow analysis of data with restricted access within one year from the request, according to extant regulations. The individual-level raw data for the replication NIMH PET cohort, the LIBD fMRI discovery cohort, and the UNIBA fMRI replication cohort are not currently publicly available because the regulation at the time of consent acquisition required participants to explicitly consent to sharing data with select institutions and impedes data sharing unbeknown to participants. The SCZ GWAS summary statistics used in this study are publicly available at: https://figshare.com/articles/dataset/scz2022/19426775. GRCh38 human genome reference genome is available here: https://ftp.ebi.ac.uk/pub/databases/gencode/Gencode_human/release_25/GRCh38.p7.genome.fa.gz. The GENCODE release 25, GRCh38.p7 annotation file used in this study is available at: https://ftp.ebi.ac.uk/pub/databases/gencode/Gencode_human/release_25/gencode.v25.basic.annotation.gtf.gz. SDA output data are available in the Supplementary Data 1. Components summary information is available in the Supplementary Data 2. GO enrichment results are available in Supplementary Data 3. The aggregated deidentified PET and fMRI data along with PRSs and technical covariates for all discovery and replication cohorts used in this study are available in Supplementary Data 4. To ensure the replicability of the results reported in this work, all processed and aggregated data (Supplementary Data 1–4), SDA input data, and SNPs used to compute C80-PRSs are also available at: https://doi.org/10.5281/zenodo.10699265. Source data are provided in this paper.

## Code availability

SDA software is publicly available at: https://jmarchini.org/software/#sda and no customization of the source code was applied. The scripts used for the analyses conducted in this study are available in the Supplementary Software file.

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

## Acknowledgements

This research was supported by the Intramural Research Program of the National Institute of Health through project ZIAMH002942; NCT00004571/ NCT00024622/ NCT00001486. This project has received funding from the European Union's Horizon 2020 research and innovation program under the Marie Skłodowska-Curie grant agreement no. 798181 awarded to G.P. (PI), and A.B., D.R.W. NIH grant 5R21MH117432-02, awarded to D.R.W. (PI), G.P. and A.B., partially supported M.P. This research has been partially supported by the project "Dopamine-dysbindin genetic interaction: a multidisciplinary approach to characterize cognitive phenotypes of schizophrenia and develop personalized treatments" (PRIN: Progetti di Ricerca di Rilevante Interesse Nazionale–Bando 2017 Prot. 2017K2NEF4) awarded to G.P., the funding initiative Horizon Europe Seeds 2021 (Next Generation EU-MUR D.M. 737/ 2021) for the project S68 CUP H99J21017550006 to G.P., from the European Union funding within the MUR PNRR Extended Partnership initiative on Neuroscience and Neuropharmacology (Project no. PE00000006 CUP H93C22000660006 "MNESYS, A multiscale integrated approach to the study of the nervous system in health and disease") to AB, AR, and GP, and the Apulian regional government for the project: "Early Identification of Psychosis Risk" to A.B. The LIBD supported tissue collection and maintenance, analysis, infrastructure, and personnel. We are grateful for the contributions of the Office of the Chief Medical Examiner of the State of Maryland, Office of the Chief Medical Examiner of Kalamazoo County Michigan, Office of the Chief Medical Examiner University of North Dakota School of Medicine, Gift of Life of Michigan, Office of the Chief Medical Examiner of Santa Clara County California, and Medical University of Sofia, Bulgaria in assisting the Lieber Institute for Brain Development in the acquisition and curation of brain tissue donations for this study. All research at the Lieber Institute for Brain Development is made possible by generous gifts from the families of Steve and Connie Lieber and Milton and Tamar Maltz. We would like to thank all the family members of the donors for their exceptional contribution. We would like to acknowledge Dr. Peter Herscovitch and the staff of the NIH PET Center for their data acquisition support. We are appreciative of all the neuroimaging research volunteers for their generous participation. We are in debt to A. Jaffe who has contributed to this work by offering data and software. We are grateful to Dr. Christopher J. Borcuk, Dr. Pasquale Di Carlo, Dr. Piergiuseppe Di Palo, Andrea Gaudio, Gianluca C. Kikidis, Ciro Mazza, Dr. Marco Papalino, and Prof. Paolo Taurisano for data collection, exploratory analyses and exchanges of ideas that have influenced this work.

## Author contributions

Conceptualization: L.S., G.P., and D.R.W. Data curation: L.S., D.E., R.P., E.D., L.A.A., J.S.B., C.F.Z., J.H.S., O.D.H. Formal analysis: L.S., D.E., R.P., G.P. Funding acquisition: G.P., D.R.W., and A.B. Investigation: L.S., D.E., G.P., C.F.Z., A.B., K.B., D.R.W. Methodology: L.S., D.E., R.P., L.A.A., J.H.S., M.P., Q.C. Project administration: G.P. Resources: J.S.B., A.G., L.A.A., M.D.G., K.G., T.H., J.K., A.F.P., T.P., A.R., M.V., W.U. Software: L.S., D.E. and R.P. Supervision: G.P. and D.R.W. Visualization: L.S., D.E. and R.P. Writing (original draft): L.S, D.E., R.P., G.P., and D.R.W. Writing (review and editing): all authors.

## Competing interests

E.D. and G.P. received lecture fees from Lundbeck. A.R. received travel fees from Lundbeck. A.B. received consulting fees from Biogen and lecture fees from Otsuka, Janssen, and Lundbeck. O.D.H. has received investigator-initiated research funding from and/or participated in advisory/speaker meetings organized by Angelini, Autifony, Biogen, Boehringer Ingelheim, Eli Lilly, Heptares, Global Medical Education, Invicro, Janssen, Lundbeck, Neurocrine, Otsuka, Sunovion, Recordati, Roche and Viatris/Mylan and was a part-time employee of H Lundbeck A/s. O.D.H. and M.V. have a patent for the use of dopaminergic imaging. D.R.W. serves on the Scientific Advisory Boards of Sage Therapeutics and Pasithea Therapeutics. The remaining authors declare no competing interests.
