## [Peer Review File · Nature Communications]

Dopamine signaling enriched striatal gene set predicts striatal dopamine synthesis and physiological activity in vivoReviewer #1 (Remarks to the Author):

This manuscript uses recently developed computational tools to dissect risk for schizophrenia into co-expression pathways. The authors identify a gene set expressed in the human striatum in postmortem brains associated with the diagnosis of schizophrenia and genetic risk for the disorder. They show that this gene set is enriched for dopamine transmission and predicts dopamine synthesis capacity measured via PET and reward activity measured by fMRI in the human striatum in living subjects. Findings in each of the examined modalities are replicated. This work is timely, the methods are rigorous, and the clinical significance of parsing genetic risk for schizophrenia into a number of gene sets, each associated with different phenotypes, is tangible. The article reads very well; therefore, I have only a few minor suggestions for the authors to consider:

1. Co-expression components were filtered for diagnosis association and then selected for genetic risk association. I agree that this is a logical procedure. However, it would be interesting to know whether other components are associated with genetic risk. Do these components also predict dopamine synthesis? This analysis would help establish the specificity of C80 or reveal other components of interest.

2. In the same vein, I wonder which components are more associated with DLPFC and hippocampus. Considering that dopamine receptors might not be detected with RNAseq in these areas, are there other pathways of interest in which schizophrenia risk converges? For example, glutamate in the DLPFC, or GABAergic transmission.

3. The authors mention that this component is also enriched for major depression and ADHD genes. Are they the same genes associated with schizophrenia, or different genes converging in the same set? This information might be relevant to compare the genetic signature with treatments.

4. There was no agreement between LIBD and GTEx data at the level of D2 isoform association with C80.

Are there differences between datasets that could explain this negative finding? For example, the age of the subjects or RNAseq coverage?

Reviewer #2 (Remarks to the Author):

This is an interesting study using a co-expression detection method called sparse decomposition of arrays (SDA) to identify, via human RNA sequencing data, genes co-expressed in post-mortem brain regions of neurotypical individuals. Authors identify a caudate nucleus predominant gene set with dopaminergic selectivity enriched for genes associated with clinical state and SCZ risk. The gene set associated with risk predicted measures of in vivo dopamine synthesis in three different cohorts of patients and comparison groups acquired at NIMH and at Kings College London and reward task related striatal activation in two different fMRI studies of healthy volunteers. Combining all these different methodologies and leveraging the availability of relevant data in humans is novel and powerful. The results are surprisingly supportive of the role of striatal dopamine function in reward functions in general and in schizophrenia related psychosis risk associated with striatal dopamine dysfunction.

I find this report noteworthy and intriguing and definitely worthy of publication in a high-profile journal.

My only concerns relate to some of the details of what the authors call replication across cohorts. For example, for the fMRI studies the first cohort identifies a region within the right caudate nucleus, and replicated this in a cluster "partially overlapping". It is not clear why the relationship would be specific to the right vs the left caudate nucleus, and if a partially overlapping cluster constitutes a replication. In addition, terms such as caudate nucleus, dorso-medial striatum, associative striatum, are used throughout the text across the different cohorts without specifying exactly how do these terms correspond to each other, and how much overlap they represent. Nevertheless, there is sufficient correspondence here to provide strong evidence in support of the conclusions made in the paper. The discussion is very thoughtful and contributes insights into the

significance of the observations.

Minor point : In abstract GTEX should be spelled out.

RE: NCOMMS-23-41404

We sincerely appreciate the thorough feedback provided by the reviewers on our manuscript. In response to their comments and suggestions, we have edited our manuscript, greatly enhancing the interpretability and the overall quality of our work. Below, we have listed the reviewers' comments in their entirety and responded to each comment point-by-point. Responses to reviewers' comments in this letter are marked in bold, with corresponding changes in the main manuscript in italics. In the main manuscript and supplementary materials corresponding changes are marked in red.

Reviewer Comments:

Reviewer #1 (Remarks to the Author):

This manuscript uses recently developed computational tools to dissect risk for schizophrenia into co-expression pathways. The authors identify a gene set expressed in the human striatum in postmortem brains associated with the diagnosis of schizophrenia and genetic risk for the disorder. They show that this gene set is enriched for dopamine transmission and predicts dopamine synthesis capacity measured via PET and reward activity measured by fMRI in the human striatum in living subjects. Findings in each of the examined modalities are replicated. This work is timely, the methods are rigorous, and the clinical significance of parsing genetic risk for schizophrenia into a number of gene sets, each associated with different phenotypes, is tangible. The article reads very well; therefore, I have only a few minor suggestions for the authors to consider:

1. Co-expression components were filtered for diagnosis association and then selected for genetic risk association. I agree that this is a logical procedure. However, it would be interesting to know whether other components are associated with genetic risk. Do these components also predict dopamine synthesis? This analysis would help establish the specificity of C80 or reveal other components of interest.
2. In the same vein, I wonder which components are more associated with DLPFC and hippocampus. Considering that dopamine receptors might not be detected with RNA-seq in these areas, are there other pathways of interest in which schizophrenia risk converges? For example, glutamate in the DLPFC, or GABAergic transmission.

We appreciate the reviewer's positive feedback overall and these first two insightful and – as the reviewer noted – related suggestions. We have now conducted follow-up analyses of the data to both (1) look for genetic risk and striatal dopamine associations with other components and (2) examine their biological/neuroanatomical correlates to identify other possible pathways of interest in which schizophrenia risk converges.

First, to determine whether other components are associated with SCZ genetic risk, we explored the association of the 69 identified components with the SCZ Polygenic Risk Score (PRS) using the same multiple linear regression methods as reported in the manuscript (see *Online methods*). We then looked at their biological characterization through enrichment and over-representation analyses as previously done (Supplementary Figure 1) to identify enrichment for SCZ risk genes as well as alternative pathways of risk convergence. Finally, we computed PRSs stratified for genes within each component. We evaluated their association with striatal dopamine synthesis capacity as measured by [¹⁸F]-FDOPA specific uptake in our discovery NC and SCZ cohorts. We employed the same methodology as described in the manuscript for C80 (see *Online methods*).

We found six additional components out of 69 associated with SCZ PRS at the nominal significance level (two-tailed $\alpha=.05$; Figure R1a). None showed significant effects of diagnosis in postmortem samples. Of these, only one component (C102) was also enriched for SCZ risk genes (empirical $p <.05$; Figure R1b). Interestingly, this component was ascribed to the hippocampus with specificity for glutamatergic synapses of hippocampal CA pyramidal and dentate gyrus neurons, among others (Figure R1b-d). None of the six SCZ-PRS-associated components were significantly associated with striatal dopamine synthesis (Figure R1e). SCZ risk variants proximal to C80 genes may thus be more associated with striatal dopamine than those in proximity to genes in other candidate risk components. Additionally, in order to disentangle the specificity of the effects of C80-PRS on brain activity, we investigated the association of the six stratified PRSs with BOLD signal on the fMRI discovery cohort as previously described (see *Online methods*). Indeed, we observed no significant association with either reward anticipation or reward consumption, further supporting the specificity of the C80-PRS for reward-related brain activity.

We have now reported these analyses in the Supplementary Notes (Page 6) as follows: *“We explored the association of the 69 identified components with SCZ PRS using the same multiple linear regression methods as reported in the main text (see Online methods). We then looked at their biological characterization through enrichment and over-representation analyses to identify enrichment for SCZ risk genes as well as alternative pathways of risk convergence (see Online methods for details). We found six additional components out of 69 associated with SCZ PRS at the nominal significance level (two-tailed $\alpha=.05$, uncorrected; Supplementary Fig. 6a). None showed significant effects of diagnosis in postmortem samples. Of these, only one component (C102) was also enriched for SCZ risk genes (empirical $p <.05$; Supplementary Fig. 1a). C102 also showed enrichment for MDD, ASD, and PTSD risk genes as well as for SCZ differentially expressed genes (DEGs) previously observed in the CN, DLPFC, and HP (all empirical $p <.05$; Supplementary Fig. 1a). The tissue loading matrix revealed the C102 component to be most active in the HP (Supplementary Figure 1a). Accordingly, cell specificity analysis showed enrichment for glutamatergic synapses of hippocampal CA pyramidal and dentate gyrus neurons, among others (Supplementary Fig. 6b).*

Consistently, KEGG pathway analysis for C102 showed enrichment for glutamatergic synapse as well as for nicotine and morphine addiction-related genes (Supplementary Fig. 6c). Finally, we computed PRSs stratified for genes within each SCZ-PRS-associated component and evaluated their association with striatal dopamine synthesis capacity as measured by [¹⁸F]-FDOPA specific uptake in our discovery NC and SCZ cohorts (see Online methods). Notably, none of the six components were significantly associated with striatal dopamine synthesis (Supplementary Fig. 6d). Additionally, in order to disentangle the specificity of the effects of C80-PRS on brain activity, we investigated the association of the six stratified PRSs with BOLD signal on the fMRI discovery cohort as previously described (see Online methods). We observed no significant association with either reward anticipation or reward consumption (results not shown), further supporting the specificity of the C80-PRS for reward-related brain activity.”

In the main text, in the Results section we refer to these analyses as follows at Page 9: “Further exploratory genetic risk association and biological characterization analyses of the other identified components are reported in the Supplementary Notes and Supplementary Fig. 1a and 6. The broader landscape of co-expression components highlights further potential pathways of interest linked to SCZ risk.” and at Page 13: “Importantly, the neuroimaging associations we identified with C80 were not significant for the other components associated with genetic risk for SCZ (see Supplementary Notes and Supplementary Fig. 6d).”

Figure R1. a, Scatter plots show SDA component scores as a function of polygenic risk for schizophrenia and include regression fit line with a shaded 95% confidence interval. Nominal two-tailed p-values are shown. b, Gene enrichment analysis results are shown for the six PRS-associated components. Orange grids show enrichment results for SCZ, other psychiatric illnesses, and immune condition GWAS risk genes. Enrichment testing results are shown for differentially expressed genes, differentially methylated genes, and loss of function variant intolerant genes in the green grid. The final lightblue grid shows component tissue specificity as determined by the tissue scores generated during the SDA process and reflects the relative contribution of component gene networks within each of the sampled regions to the overall component. Adjusted p-values shown are empirical

p-values obtained from permutation tests. **c**, Heatmaps show cell-type marker genes overrepresentation using human (lightblue) and mouse (brown) single-cell atlases. FDR-adjusted *p*-values are shown. **d**, KEGG enrichment of all six PRS-associated components. FDR-adjusted *p*-values are reported. **e**, Forest plots show metaanalyses of the association between the six stratified PRSs with whole-striatum dopamine synthesis capacity in the PET discovery cohort. 99.5% confidence intervals are shown.

Abbreviations: ADHD: attention deficit hyperactivity disorder; ASC: astrocytes; ASD: autism spectrum disorder; BD: bipolar disorder; CD: Crohn's disease; CN: Caudate Nucleus; DEGs: differentially expressed genes; DLPPFC: dorsolateral prefrontal cortex; DMGs: differentially methylated genes; END: endothelial cells; HP: hippocampus; exCA: pyramidal neurons from the hippocampal CA region; exDG: granule neurons from the hippocampal dentate gyrus; exPFC: pyramidal neurons from the prefrontal cortex; GABA: GABAergic interneurons; LoF: loss of function intolerant genes; MDD: major depressive disorder; MG: microglia; NC: Neurotypical controls; NSC: neuronal stem cells; OCD: obsessive compulsive disorder; ODC: oligodendrocytes; OPC: oligodendrocyte precursor cells; PRS: polygenic risk score as reported by the third wave (primary) analyses of the Psychiatric Genetics Consortium2; PTSD: posttraumatic stress disorder; SA: suicide attempt; SCZ: Patients with schizophrenia and UC: ulcerative colitis.

3. The authors mention that this component is also enriched for major depression and ADHD genes. Are they the same genes associated with schizophrenia, or different genes converging in the same set? This information might be relevant to compare the genetic signature with treatments.

The Venn diagram below shows the intersection between these gene sets (Figure R2). Indeed, the majority of C80 genes enriched for three disorders are not overlapping, supporting the reviewer's second hypothesis that different risk genes converge in the same set.

We have now added the Venn diagram in Figure 2c and reported it in the Results section (Page 8) as follows: “*Biological characterization of this component showed enrichment for SCZ, major depressive disorder (MDD) and attention deficit hyperactivity disorder (ADHD) risk genes (Fig. 2).*”

Figure R2. Venn diagram showing intersection between C80 genes and SCZ, MDD and ADHD GWAS risk genes. Blank regions indicate no common genes. In the case of a single gene result, that gene is listed.

4. There was no agreement between LIBD and GTEX data at the level of D2 isoform association with C80. Are there differences between datasets that could explain this negative finding? For example, the age of the subjects or RNAseq coverage?

We appreciate this reviewer's insightful comment, which led us to reevaluate the discrepancies between the datasets. It is important to acknowledge that our LIBD discovery dataset encompasses both neurotypical control (NC) and SCZ samples, featuring a broad

age range from 0 to 96 years, including specific younger age groups. This dataset is also characterized by a diverse ethnic composition, with 55% of African American and the remainder of European ancestry, as shown in Table 1. In contrast, the GTEx replication cohort is more homogeneous, consisting solely of NC samples of European ancestry aged between 20 and 79 years, with a mean age significantly older than LIBD (GTEx: 59.2 ± 9.6 years; LIBD: 46.2 ± 16.9 years; Wilcoxon rank sum test: one-tailed p-value $< 2.2 \times 10^{-16}$). Furthermore, as already indicated in the limitations section of our paper, the GTEx sample size is notably smaller than that of the LIBD. This difference could potentially affect the precision of our decomposition approach.

Regarding RNAseq coverage, the LIBD dataset utilized the Illumina HiSeq 3000 platform, generating a mean of 125.2 million 100-bp paired-end reads per sample (Collado-Torres, Burke et al. 2019, Benjamin, Chen et al. 2022). On the other hand, the GTEx project employed the HiSeq 2000 or HiSeq 2500 systems, aiming for a coverage of 50 million reads, with a median of approximately 82 million total reads achieved (GTEx portal: <https://www.gtexportal.org/home/methods#AboutSamples>). These figures suggest a greater resolution of LIBD data to detect transcript-level information.

Despite these distinctions in demographic and technical aspects, at the total gene level, our tensor decomposition analysis successfully identified consistent co-expression patterns related to dopaminergic neurotransmission across both datasets. These unavoidable dataset discrepancies may hinder replications at transcript-level.

We have now amended our Discussion to provide a comprehensive understanding of the study's context and implications in the limitation section (Page 18): *“Limitations of this study include the relatively small sample size used in the gene co-expression analysis, which is critical for decomposition approaches. To obtain the 3D tensor used as input for SDA, we had to exclude samples without available data in all three brain regions analyzed. This filter was especially limiting for the GTEx replication dataset. It is also important to acknowledge unavoidable dataset discrepancies that might have hindered replications in the DRD2 transcript-level analysis. Table 1 delineates the demographic heterogeneity of LIBD samples in contrast with the more homogenous GTEx cohort, especially in age distributions (GTEx: 59.2 ± 9.6 years; LIBD: 46.2 ± 16.9 years; Wilcoxon rank sum test: one-tailed p-value $< 2.2 \times 10^{-16}$). Additionally, the substantial difference in sequencing depth (LIBD: mean of 125.2 million reads per sample; GTEx: mean of 50 million) suggests a different resolution across the two datasets in detecting fine-grained transcript-level variations. Despite these distinctions in demographic and technical aspects, at the total gene level, our tensor decomposition analysis successfully identified consistent co-expression patterns related to dopaminergic neurotransmission across both datasets. These results suggest that the finer biological resolution required for DRD2 isoform association might be more sensitive to the variations inherent in the datasets.”*

Reviewer #2 (Remarks to the Author):

This is an interesting study using a co-expression detection method called sparse decomposition of arrays (SDA) to identify, via human RNA sequencing data, genes co-expressed in post-mortem brain regions of neurotypical individuals. Authors identify a caudate nucleus predominant gene set with dopaminergic selectivity enriched for genes associated with clinical state and SCZ risk. The gene set associated with risk predicted measures of in vivo dopamine synthesis in three different cohorts of patients and comparison groups acquired at NIMH and at Kings College London and reward task related striatal activation in two different fMRI studies of healthy volunteers. Combining all these different methodologies and leveraging the availability of relevant data in humans is novel and powerful. The results are surprisingly supportive of the role of striatal dopamine function in reward functions in general and in schizophrenia related psychosis risk associated with striatal dopamine dysfunction.

I find this report noteworthy and intriguing and definitely worthy of publication in a high-profile journal.

We would like to express our gratitude to the reviewer for the constructive revision of our manuscript, and we sincerely appreciate the positive feedback on the findings of our work. Below, we have addressed each highlighted issue raised by the reviewer.

My only concerns relate to some of the details of what the authors call replication across cohorts. For example, for the fMRI studies the first cohort identifies a region within the right caudate nucleus and replicated this in a cluster “partially overlapping”. It is not clear why the relationship would be specific to the right vs the left caudate nucleus, and if a partially overlapping cluster constitutes a replication.

We thank the reviewer for this insightful comment. In the revised manuscript, we provided a more detailed interpretation of the results of the fMRI study across the two independent fMRI cohorts. In general, in the presence of interindividual differences in brain anatomy and signal localization, it is reasonable to expect variations in signal localization at the group level across different cohorts (Seghier 2008). To quantitatively address the reviewer’s question of subregional overlap, we evaluated the distribution of the significant striatal clusters observed in the discovery and replication cohorts using canonical functional striatum regions of interest (ROI) according to the parcellation described by Mawlawi, Martinez et al. (2001), which defined three subregions—associative, limbic, and sensorimotor (Figure R3a)—that underlie distinct functions of the striatum based on the cortical afferents each of these sub-regions projects or receives. We calculated the percentage of clusters where we independently found a significant TFCE-FDR<0.05 effect of the C80-PRS, overlapping with the three striatal subregions, excluding voxels outside the gray matter. Both clusters predominantly fell within the associative striatum (Discovery=95%; replication=99%) with a minimal percentage in the limbic striatum in the discovery cohort

(5%) and sensorimotor striatum in the replication cohort (1%). These findings suggests that the effect is convincingly related to activity in the same striatal sub-region encompassing the caudate nucleus (Figure R3b). Notably, McCutcheon, Beck et al. (2018) also localized the locus of dopaminergic dysfunction in schizophrenia in the associative striatum.

Figure R3. a. sections showing the localization of the right striatum ROIs divided in associative, limbic, and sensorimotor sub-regions depicted in blue, red, and green respectively. b. Pie charts depicting the percentage of fMRI voxels associated with the C80 Parsed Polygenic Risk Score at the voxel-wise level in the discovery and replication samples (TFCE-FDR<.05) within each of the striatal sub-divisions. c. Zooming section showing the overlap extension of the clusters significantly associated with the C80 Parsed Polygenic Risk Score at the voxel-wise level in the discovery (yellow) and replication (blue) samples (TFCE-FDR<.05) covering 6 voxels (162 mm³) within the right associative striatum ROI.

Furthermore, we have now quantified the extent of voxelwise overlap between the two independently detected clusters from our discovery and replication fMRI analyses, which corresponds to 162 mm³. The overlap is in the head of the caudate nucleus, the same region used in the postmortem study. The rest of these two overlapping clusters is still contained in the associative striatum as defined by Mawlawi, Martinez et al. (2001). The spatial correlation of fMRI signals may blur the accuracy of signal localization across different scanners and experiments. The results of spatial overlap between the two clusters have been reported in the main text, while the additional parceling has been reported in the Supplementary Notes (Page 9).

Based on these considerations, we have edited the Results section (Pages 12-13) as follows: “Finally, we investigated the association of C80-PRS with striatal functioning using fMRI in participants who performed a reward processing task (see Online methods and Supplementary Notes for details). We found that the C80-PRS1 was positively correlated with differential anticipatory activation in the right associative striatum during high vs low motivation assessed in a discovery sample of 86 NC (Table 1; C80-PRS1: $p[TFCE-FDR]=.04$; $Z=3.54$; $x=17$; $y=15$; $z=-5$; 60 voxels). Specifically, participants with higher C80-PRS, and thus higher predicted striatal dopamine synthesis capacity, showed greater striatal activation when they expected a reward during the task (Fig. 4b). We consistently found this association in an additional independent sample of 55 NC (Table 1; C80-PRS1: $p[TFCE-FDR]=.03$; $Z=3.75$; $x=18$; $y=17$; $z=5$; 34 voxels) in a cluster located once again in the associative striatum (voxel-wise discovery-replication overlap volume: 162 mm³; see Supplementary Notes and Supplementary Fig. 7).”

Furthermore, we detailed the localization of the effects of C80-PRS on reward-related brain activity in the Discussion section (Page 16), as follows: “Accordingly, relative dopamine depletion attenuates striatal activation during the same task in healthy subjects. We found convergent positive associations between SCZ risk within the dopamine-system-enriched C80 gene set and anticipatory activation in both our discovery and replication cohorts. The activation clusters localized to the head of the caudate, the same region used in the postmortem study. Moreover, our results are consistent with past findings of positive correlations between polygenic risk for SCZ and striatal activation during the MID task in a large sample of healthy adolescents.”.

To disambiguate the terminology used to report the effects across the independent fMRI cohorts, we have rephrased as follows in the Discussion (Page 19): “*Finally, it is possible that sample size limitations in the fMRI dataset coupled with the substantial variability in the degree and direction of laterality across individuals might have affected the localization of the effects and prevented the identification of weaker but important bilateral activations at the voxel-wise level, although the consistency of findings across independent cohorts and at the ROI level mitigates this concern.*”

Regarding the laterality of the effect, we did not have any *a priori* hypotheses about the laterality of the C80-PRS's effect, considering that: (i) although an abnormal left lateralization of presynaptic dopamine density has been reported in SCZ patients (Hietala, Syvälahti et al. 1995, McGowan, Lawrence et al. 2004), no such asymmetries were observed in neurotypical individuals (Vollenweider, Vontobel et al. 1999); (ii) the striatum has been bilaterally associated with reward processing (Knutson, Adams et al. 2001) in both neurotypicals (Arsalidou, Vijayarajah et al. 2020) and patients with SCZ (Leroy et al. 2015); (iii) there is no strong evidence in the literature to suggest an *a priori* hypothesis for which an increased lateralized striatal dopamine synthesis should correlate with a striatal activation during the monetary incentive delay task with the same lateralization, let alone in correlation with genetic scores. As for the location of the effects, there may be considerable variability in the degree and direction of laterality among individuals. Some individuals may exhibit strong lateralization of certain functions, while others may show more bilateral activation (Seghier 2008).

Of course, voxel-wise analyses are subject to statistical thresholding that may mask relatively weaker effects, thus emphasizing the impression of lateralized signals, even when hemispheric differences are actually small. We thus asked whether an effect similar to the one detected in the right hemisphere could be spotted in the left striatum. We extracted the individual signal from striatum ROIs using an uncorrected threshold of $\alpha < 0.005$ (Sacchet and Knutson 2013). We employed the six striatum ROIs (right and left associative, sensorimotor, and limbic subregions per Mawlawi, Martinez et al. (2001) and as described above). Subsequently, we performed associations of the C80-PRS with the signal extracted from the individual activation maps from the left and the right ROIs. Given that the uncorrected signal derived from the ROIs may be susceptible to type-I errors, *post-hoc* associations were adjusted for multiple comparisons to account for the number of tests conducted ($k=6$). Extracting the averaged signal from the associative striatum ROIs, we confirmed the significant positive associations on the signal extracted from the right associative striatum ROI (discovery: $t(84)=3.34$; $pFDR=0.006$; replication: $t(53)=3.45$; $pFDR=0.005$) along with significant positive associations on the signal extracted from the left associative striatum ROI, though with smaller effect sizes (discovery: $t(84)=2.39$; $pFDR=0.05$; replication: $t(53)=3.28$; $pFDR=0.009$; Figure R4). No associations were significant in the limbic and sensorimotor striatum ($pFDR > 0.05$) in either sample. This

additional evidence shows a consistent activation pattern in both the left and right associative striatum (same mask as the PET study). Nevertheless, the potential influence of individual variability in signal localization, coupled with the limited sample sizes could have prevented the identification of under-threshold BOLD effects at the voxel-wise level, as we have now stated in the Discussion section (Page 19):

“Finally, it is possible that sample size limitations in the fMRI dataset coupled with the substantial variability in the degree and direction of laterality across individuals might have affected the localization of the effects and prevented the identification of weaker but important bilateral activations at the voxel-wise level, although the consistency of findings across independent cohorts and at the ROI level mitigates this concern”.

For completeness, the additional ROI analyses on the left striatal activation and association with C80-PRS have been included in the Supplementary Notes (Page 10) and referred to in the Results section (Page 13) as follows: *“It is worth mentioning that we confirmed the C80-PRS effects also on the BOLD signal extracted from both the right and left associative striatum ROIs (see Supplementary Notes for details).”.*

Figure R4. a. Sections showing the activation patterns at the group level within the bilateral striatum ROIs (Mawlawi, Martinez et al. 2001, McCutcheon, Beck et al. 2018) at $p < 0.005$, $k=20$ in the discovery and replication samples. b. Scatterplot showing the associations between the signal extracted from the individual activation maps from the right associative striatum ROI and the C80 parsed Polygenic Risk Score.

In addition, terms such as caudate nucleus, dorso-medial striatum, associative striatum, are used throughout the text across the different cohorts without specifying exactly how do these terms correspond to each other, and how much overlap they represent.

We appreciate the reviewer for bringing to our attention the inconsistency in the terms used to describe the results in the different PET and fMRI cohorts. For the sake of clarity, we have now standardized the terminology used in the neuroimaging analyses by referring to the spatial localization performed for all PET and fMRI results across cohorts using the ROIs derived from Mawlawi, Martinez et al. (2001) and as described in McCutcheon, Beck et al. (2018), i.e. associative, sensorimotor and limbic striatum (see Figure R1a). Meanwhile, in the post-mortem analyses, we have retained "caudate nucleus" in reference to the histological section on which the RNA-sequencing was performed, although the head of the caudate that was used in dissections is effectively part of the associative striatum.

Nevertheless, there is sufficient correspondence here to provide strong evidence in support of the conclusions made in the paper. The discussion is very thoughtful and contributes insights into the significance of the observations.

Minor point: In abstract GTEX should be spelled out.

We have now spelled out “GTE_x” as “Genotype-Tissue Expression” in the abstract, as suggested.

Again, many thanks for the thoughtful review of our manuscript.

References

- Arsalidou, M., S. Vijayarajah and M. Sharaev (2020). "Basal ganglia lateralization in different types of reward." Brain Imaging Behav **14**(6): 2618-2646.
- Benjamin, K. J. M., Q. Chen, A. E. Jaffe, J. M. Stolz, L. Collado-Torres, L. A. Huuki-Myers, E. E. Burke, R. Arora, A. S. Feltrin, A. R. Barbosa, E. Radulescu, G. Pergola, J. H. Shin, W. S. Ulrich, A. Deep-Soboslay, R. Tao, M. Matsumoto, T. Saito, K. Tajinda, D. J. Hoepfner, D. A. Collier, K. Malki, B. B. Miller, M. Furey, D. Hibar, H. Kolb, M. Didriksen, L. Folkersen, T. Kam-Thong, D. Malhotra, J. H. Shin, A. E. Jaffe, R. Narurkar, R. E. Straub, T. M. Hyde, J. E. Kleinman, D. R. Weinberger, T. M. Hyde, J. E. Kleinman, J. A. Erwin, D. R. Weinberger, A. C. M. Paquola and C. the BrainSeq (2022). "Analysis of the caudate nucleus transcriptome in individuals with schizophrenia highlights effects of antipsychotics and new risk genes." Nature Neuroscience **25**(11): 1559-1568.
- Collado-Torres, L., E. E. Burke, A. Peterson, J. Shin, R. E. Straub, A. Rajpurohit, S. A. Semick, W. S. Ulrich, A. J. Price, C. Valencia, R. Tao, A. Deep-Soboslay, T. M. Hyde, J. E. Kleinman, D. R. Weinberger and A. E. Jaffe (2019). "Regional Heterogeneity in Gene Expression, Regulation, and Coherence in the Frontal Cortex and Hippocampus across Development and Schizophrenia." Neuron **103**(2): 203-216.e208.
- Hietala, J., E. Syvälahti, K. Vuorio, V. Rökköläinen, J. Bergman, M. Haaparanta, O. Solin, M. Kuoppamäki, O. Kirvelä, U. Ruotsalainen and et al. (1995). "Presynaptic dopamine function in striatum of neuroleptic-naive schizophrenic patients." Lancet **346**(8983): 1130-1131.
- Knutson, B., C. M. Adams, G. W. Fong and D. Hommer (2001). "Anticipation of increasing monetary reward selectively recruits nucleus accumbens." J Neurosci **21**(16): Rc159.
- Mawlawi, O., D. Martinez, M. Slifstein, A. Broft, R. Chatterjee, D. R. Hwang, Y. Huang, N. Simpson, K. Ngo, R. Van Heertum and M. Laruelle (2001). "Imaging human mesolimbic dopamine transmission with positron emission tomography: I. Accuracy and precision of D(2) receptor parameter measurements in ventral striatum." J Cereb Blood Flow Metab **21**(9): 1034-1057.
- McCutcheon, R., K. Beck, S. Jauhar and O. D. Howes (2018). "Defining the Locus of Dopaminergic Dysfunction in Schizophrenia: A Meta-analysis and Test of the Mesolimbic Hypothesis." Schizophr Bull **44**(6): 1301-1311.
- McGowan, S., A. D. Lawrence, T. Sales, D. Quested and P. Grasby (2004). "Presynaptic dopaminergic dysfunction in schizophrenia: a positron emission tomographic [¹⁸F]fluorodopa study." Arch Gen Psychiatry **61**(2): 134-142.
- Sacchet, M. D. and B. Knutson (2013). "Spatial smoothing systematically biases the localization of reward-related brain activity." Neuroimage **66**: 270-277.
- Seghier, M. L. (2008). "Laterality index in functional MRI: methodological issues." Magnetic Resonance Imaging **26**(5): 594-601.
- Vollenweider, F. X., P. Vontobel, D. Hell and K. L. Leenders (1999). "5-HT modulation of dopamine release in basal ganglia in psilocybin-induced psychosis in man--a PET study with [¹¹C]raclopride." Neuropsychopharmacology **20**(5): 424-433.

Reviewer #1 (Remarks to the Author):

The authors adequately answered all the questions raised by this referee

Reviewer #2 (Remarks to the Author):

The authors addressed all comments thoughtfully and extensively. I have no further concerns, and I look forward to seeing this paper published.